# Ice Binding Proteins: Diverse Biological Roles and Applications in Different Types of Industry

**DOI:** 10.3390/biom10020274

**Published:** 2020-02-11

**Authors:** Aneta Białkowska, Edyta Majewska, Aleksandra Olczak, Aleksandra Twarda-Clapa

**Affiliations:** Institute of Molecular and Industrial Biotechnology, Lodz University of Technology, Stefanowskiego 4/10, 90-924 Łódź, Poland; edyta.majewska@dokt.p.lodz.pl (E.M.); aleksandra.twarda-clapa@p.lodz.pl (A.T.-C.)

**Keywords:** ice binding proteins, cryopreservation, antifreeze proteins

## Abstract

More than 80% of Earth’s surface is exposed periodically or continuously to temperatures below 5 °C. Organisms that can live in these areas are called psychrophilic or psychrotolerant. They have evolved many adaptations that allow them to survive low temperatures. One of the most interesting modifications is production of specific substances that prevent living organisms from freezing. Psychrophiles can synthesize special peptides and proteins that modulate the growth of ice crystals and are generally called ice binding proteins (IBPs). Among them, antifreeze proteins (AFPs) inhibit the formation of large ice grains inside the cells that may damage cellular organelles or cause cell death. AFPs, with their unique properties of thermal hysteresis (TH) and ice recrystallization inhibition (IRI), have become one of the promising tools in industrial applications like cryobiology, food storage, and others. Attention of the industry was also caught by another group of IBPs exhibiting a different activity—ice-nucleating proteins (INPs). This review summarizes the current state of art and possible utilizations of the large group of IBPs.

## 1. Introduction

Current knowledge about living organisms states that there is no place on the surface of the Earth, or in the outer crust [1], ocean crust [2], or in the clouds [3], where life does not exist. Over 80% of Earth’s surface is exposed to temperatures below 5 °C during the year and includes the areas of the North Pole and Antarctica, deep sea, permafrost, and mountain glaciers. However, despite these extreme conditions, even permafrost areas are rich in organisms called psychrophiles that belong to three different domains (Bacteria, Archaea, and Eukarya including yeast, filamentous fungi, lichens, small invertebrates, and plants) [2,4]. Recent studies have shown that psychrophiles developed a number of adaptations that enable them to overcome the effects of low temperature. Cold adaptation requires multiple changes at the molecular level such as: (1) increasing the fluidity of the lipid membrane, (2) synthesis of special proteins—chaperonins, (3) synthesis of cryoprotectants, (4) reduction of growth factors, (5) synthesis of cold-active enzymes, and (6) metabolic changes [5].

Cold-adapted microorganisms can be divided into two groups: obligate and facultative psychrophiles. Obligate psychrophiles prefer the temperature below 20 °C; their optimum growth is at ~15 °C. These organisms are found in polar regions, mountains, glaciers, and the deep sea. Facultative psychrophiles are organisms that have an optimal growth temperature above 20 °C, but can survive below 0 °C. They live in environments with periodic and seasonal temperature fluctuations. Currently, the minimum temperature for enabling reproduction is considered −12 °C and at −20 °C metabolism is stopped [6]. Water availability is a key factor that allows the growth of microorganisms at low temperatures. Besides the low temperatures and water shortage, psychrophiles often have to cope with other stressors, such as high pressure, high salinity, UV radiation, hypoxia, or low nutrient availability. The importance of cold-adapted microorganisms is not only recognized due to their unique survival mechanisms, but also because of their role in biodegradation of organic matter and circulation of essential nutrients in polar and subpolar regions [7].

The discovery of ice binding proteins (IBPs) synthesized by cold-adapted microorganisms as a strategy of overcoming low temperatures resulted in an increasing interest in their specificity. Their properties such as lowering the freezing point of water or protection from recrystallization during storage opened up a prospect of IBPs becoming useful tools in commercial applications. Special attention is paid to antifreeze proteins (AFPs), which were found to be beneficial biomolecules for the food industry [8], agriculture [9], cryosurgery, and cryopreservation of cells [10], tissues [11], and organs [12].

This review describes the biodiversity of IBPs in terms of their structures, roles, sources of origin, and mechanisms of action. The main focus of the article is concentrated on the already existing and potential industrial applications of this fascinating group of biomolecules.

## 2. Biological Roles of IBPs

### 2.1. Antifreeze Protection

Water is an essential molecule for life, but it can also lead to cell death. Freezing is lethal for most organisms. The formation of ice crystals can damage cell membranes, which results in cell disruption. Adapting to life in harsh conditions, psychrophiles have developed a number of strategies that allow them to survive. The strategies can range from behavioral (hibernation under mud, migration to warmer areas, annual lifespans) to physiological including freezing avoiding (FA) by supercooling and freezing tolerance (FT) [13]. FA strategy allows organisms to keep body fluids undercooled and to completely remove ice nucleating agents. The organisms that use this strategy usually live in the areas with constant extreme temperatures, i.e., the northern hemisphere, where there are seasonal, longer, and predictable temperature drops. The organisms adapting to this strategy, often associated with high initial and progress stress, energy cost, and the possibility of death in case of flash freezing, depress the freezing point of internal fluids to avoid ice formation. FT strategy promotes premature and controlled ice formation in extracellular spaces; intracellular freezing is lethal. Two thirds of the frozen body fluids are considered the limit of tolerance. Organisms using this approach are vulnerable to freezing and thawing damage. However, it allows the consumption of smaller amounts of energy and survival at temperatures below 5 to 40 °C than organisms using FA. FT is more common in shallow waters, terrestrial regions, and the southern hemisphere, due to the less extreme and more variable cold periods. Moreover, supercooling inhibits the formation of ice within the tissue by ice nucleation and allows the cells to maintain water in a liquid state—cell solutions are kept between the equilibrium freezing point and the homogeneous ice nucleation temperature of water (usually between −1 and −41 °C). As a consequence, the water within the cell stays separate from the extracellular ice. In contrast, tissues that avoid stress survive freezing temperatures by supercooling, a process in which cell solutions are kept between the equilibrium freezing point and the homogeneous ice nucleation temperature of water (usually between −1 and −41 °C) [13]. To sum up, the interplay between FA, FT, and utilization of supercooling is not straightforward, as FT and FA may coexist within the same system, based on different mechanisms.

These two survival strategies are realized by the synthesis of specific salts, saccharides, peptides, or proteins, IBPs, whose main role is to depress the freezing temperature and affect the shape and size of the ice crystals. The definition of IBPs is broad (Figure 1) and includes AFPs and ice-nucleating proteins (INPs, Figure 1A).

Increasing the tolerance to freezing is based on two activities of AFPs: thermal hysteresis (TH) and ice recrystallization inhibition (IRI). TH causes the decrease of the freezing point of body fluid and was first described by DeVries et al. in 1969 [14] in polar fish living in shallow waters. Thanks to the gap between melting and freezing point, psychrophilic organisms such as fishes and insects with TH up to >5 °C [15] can survive because the unlimited growth of ice crystals in body fluid is inhibited (Figure 1B). On the other hand, IRI activity preserves from freeze-thaw stress. Cold-resistant organisms during periodic temperature fluctuations around subzero temperatures are exposed to the recrystallization of ice, i.e., to the merging of ice crystals, changes in their shape, or the growth of larger ice crystals at the expense of smaller ones. Inhibition of this process prevents the freezing of liquids inside the cells, and consequently, the cell death. To sum up, the interaction of both TH and IRI activity of AFPs makes the unavoidable ice crystals small and non-lethal to cells (Figure 1B,E). The level and type of expressed AFPs depends on the type of organism, survival strategy, and the environment.

### 2.2. Other Roles

Despite the abovementioned activities of AFPs in the cell, some microorganisms also synthesize secreted AFPs to have a constant access to nutrition and water. Bacteria, yeast, fungi, and algae may live and grow in water microchannels in large sea ice created by extracellular IBPs (Figure 1C). Some Arctic bacteria (*Marinomonas primoryensis*) maintain access to oxygen and nutrients by binding to the ice surface through ice-binding domains of large Ca^2+^-dependent adhesion proteins (MpAFP, 1.5 MDa) [16,17,18]. MpAFP is an extracellular, multidomain (Repeat I–V, RI–RV), “train-like” protein that at one end anchors to bacterium and at the other (by RIV) binds to ice (Figure 1D). Moreover, the long RII provides a distance between the cell and the surface of the ice layer and prevents bacterium’s incorporation into it. Hence, the expression of MpAFP helps the strictly aerobic *M. primoryensis* remain in the upper reaches of the ice-covered lake where oxygen, nutrients, and light are most abundant [17,18]. Another extremely interesting role of IBPs is their ability to induce the nucleation of ice (Figure 1A). INPs produced by pathogenic bacteria, e.g., *Pseudomonas syringae* under conditions from −5 to −12 °C, increase the tolerance to freezing, e.g., by forming ice crystals outside the cell, thereby using the heat generated as a heat source. Furthermore, INPs can cause injuries in the epithelium of fruits and vegetables to give bacteria the access to nutrients [7]. Taken together, the expression of IBPs may have a connection with bacterial virulence.

## 3. Structural Diversity, Characteristics, and Classification of IBPs

The first discovery of AFPs was by DeVries et al. over 50 years ago in Antarctic notothenioid fishes [14]. They were present at millimolar concentrations in fish blood and prevented the growth of ice crystals. The current state of knowledge leads to the conclusion that these proteins are widespread and have been found in many others psychrophilic organisms, such as bacteria [19,20], fungi [21,22], yeast [23,24], diatoms [25], plants [26,27], and insects [28].

The major function of AFPs is the protection of organisms from freezing by increasing the freezing tolerance by ice adhesion [29]. Although AFPs of various origins have one thing in common—they all bind irreversibly to the ice surface and inhibit its growth [30]—it has been demonstrated in that they have very different three–dimensional structures (Figure 2). These structures can range from long, parallel α–helices (Figure 2A), through four-helix bundles, to large ordered β-solenoids (Figure 2G) [31]. The list of experimental structures deposited in the Protein Data Bank (PDB) is still growing. In addition, these proteins have a wide range of molecular mass, from 3.24 kDa in winter flounder (*Pseudopleuronectes americanus*) fish AFPs to 1.5 MDa in Antarctic Gram-negative bacterium, *Marinomonas primoryensis* [16,27,32]. The evolutionary pressure of living in harsh conditions resulted in the fact that similar AFPs were found in fish from the north (cods) and south (notothenioids) hemispheres [33]. Another result of the environmental pressure of adaptation to living in a subzero temperatures was the lateral transfer of the AFPs’ genes. For example, the gene of the AFP that consists of several β–solenoids with a binding α-helix has been found in microorganisms such as bacteria, fungi, yeast, diatoms, and algae [25]. Based on the amino-acid sequence and structure of AFPs, it can be also concluded that these proteins are evolutionary diverse, e.g., some fish glycoproteins were found to be derived from the trypsinogen gene, while plant AFPs are evolutionarily originated from genes of chitinases—a class of pathogenesis-related proteins. The different genes of IBPs in different organisms are the result of a combination of independent evolutionary events, from convergence through lateral gene transfer. AFPs from bacteria, fungi, insects, and plants are characterized by the β-solenoid motif in structural organization [34]. In insect and plant AFPs (Figure 2E,G), the β-solenoids are formed as a regular repeat. However, an insect snow flea expresses a significantly different protein—sfAFP consists of six antiparallel polyproline type II coils (Figure 2D) [35]. In microorganisms like bacterium *Colwellia* and yeast *Leucosporidium* (Figure 2H,I), the β-loops are irregular and of different length. Moreover, the α–helix arranged along the β-sheet axis appears in the structures [24,36]. In contrast, the RIV ice-binding domain of a large microbial adhesin MpAFP does not follow this fold and is stabilized by regularly arranged Ca^2+^ ions (Figure 2J) [17]. Also RII of MpAFP, which serves a spacer domain projecting RIV away from other cell surface molecules, is stabilized by numerous Ca^2+^ ions that strengthen and extend the massive array of RII domains to form a rigid rod-like structure (Figure 2K) [18].

The classification of AFPs was determined based on the origin of these proteins. Research distinguished proteins of fish origin divided into four subgroups, type I, type II, type III, and AFPG (antifreeze glycoprotein), and non–fish origins that include microbial proteins (derived from bacteria, diatoms, and fungi) and proteins isolated from insects and plants. Characteristic features of the particular groups are described below and summarized in Table 1.

### 3.1. AFPs from Fish

AFPG is synthesized by Antarctic notothenioids and northern cods and characterized by small molecular weight ranging from 3 to 33 kDa (Table 1). AFPGs were divided into eight subcategories, of which the AFGP1 group contains proteins with the highest molecular mass of 33.8 kDa and the AFPG8 group with the lowest (3 kDa) [37]. These proteins are constructed from 4 to 50 repeats of the triad sequence Ala–Ala–Thr, with a sugar branch (galactose *N*–acetylgalactosamine) at each hydroxyl group of the side chains of threonines. The AFGP gene derives from the recruitment and iteration of a small region spanning the boundary between the first intron and second exon of the trypsinogen gene. This new segment was expanded and then iteratively duplicated to produce 41 tandemly repeated segments [38]. Due to its structure and disordered nature, no experimental structures of these proteins have been solved so far [39].

Type I AFPs were isolated from right-eyed flounders, sculpins, and Alaskan plaices. These proteins have a small mass ranging from 3 to 4.5 kDa, with the exception of AFP “Maxi” (PDB 4KE2, Figure 2A), which has a molecular weight of ~40 kDa, exists as a dimer, and is expressed by the winter flounder [40]. Type I AFPs have a highly repeatable amino-acid structure and are made of 11 residual alanine-rich repeats forming an α-helix.

Type II proteins are characterized by a globular shape and come from American herrings, sea ravens, and smelts. In terms of structure, they are rich in cysteine and similar to lectins (sugar-binding proteins). Their molecular weight ranges from 11 to 24 kDa [41].

Type III AFPs are synthesized by eel pout, ocean pout, and wolfish. They are characterized by a globular shape, a low molecular weight of about 6–7 kDa, and no repeatability in the amino-acid sequences that do not contain cysteine residues. They can be divided into two subgroups: quaternary amino-ethyl (QAE) and sephadex-binding (SP) isoforms, based on sequence similarity and isoelectric point [42].

### 3.2. AFPs of Non-Fish Origin

#### 3.2.1. AFPs from Plants

Like fish, plants also synthesize AFPs to protect against freezing damage associated with ice crystal formation in the apoplasts. Plant AFPs have lower TH activity at the expense of IRI activity [27,43]. During periodic temperature fluctuations, this activity prevents the formation of large ice crystals that can damage cells. AFPs from plants (Table 1) were first discovered almost 30 years ago in the apoplastic extracts from winter rye (*Secale cereale*) [26]. This AFP is characterized by the TH of about 0.03 °C at 0.1 mg/mL and creates hexagonal bipiramidal ice crystals [44]. Most of plant AFPs have the low value of TH (0.1–0.5 °C), with an exception of TH activity ~2 °C from spruce (*Picea* sp.) [45]. Interestingly, plant AFPs can control ice crystal growth at micromolar concentrations (as low as 3 µg/mL for *Lolium perenne* LpAFP) [46]. Furthermore, to increase the antifreezing effect, some plants synthesize non-protein glycolipid molecules which are composed of mannose and xylose residues and enhance TH to 3 °C [47]. Most of plant AFP genes show sequence homology to pathogen-related genes (PR). These proteins have both ice binding activity and hydrolytic activity [48]. They therefore show an interesting evolutionary context of providing protection against freezing and pathogen attack. An additional anti-pathogenic role of plant AFPs is to inactivate the INPs produced by some Gram-negative pathogenic bacteria, e.g., *Pseudomonas* sp., which in this way obtain the entrance to the intracellular nutrients [49].

#### 3.2.2. AFPs from Insects

Insect proteins are so far the most active characterized AFPs (Table 1). The TH value of most insect AFPs exceeds 5 °C. Some insect AFP can decrease freezing temperature below 13 °C. Proteins showing TH activity have been found in insects such as centipedes [50], spiders [51], ticks [52], mites [53], and springtails [54,55]. Furthermore, two types of insect AFPs are distinguished: Tenebrio (found in beetles) and Dendroides (found in *Lepidoptera*), which belong to different insect families [56]. AFPs belonging to these groups are similar and have greater TH activity than other known proteins from this family. In terms of structure, they consist of 12- or 13-mer repetitions ranging from 8 to 12 kDa, where every sixth amino-acid residue is the cysteine [56]. The first identified TH activity in insects was found in hemolymph and other body fluids in the *Tenebrio molitor* beetle by Ramsay in 1971, during his study on the rectal complex [57]. At that time, these proteins were not connected with the antifreeze activity. However, analyzing the hemolymph of other insects, it was found that AFPs protect them in the winter. Moreover, 26% of the 75 insects tested in Alaska displayed TH activity [58].

#### 3.2.3. AFPs of Microbial Origin

The discovery of fish AFPs was followed by extensive search for other sources of these specific molecules motivated by the cost of purification and problems with large scale production of the recombinant fish proteins [59]. It turned out that other psychrophilic microorganisms have adapted to live in ice-filled habitats too. Organisms such as diatoms, algae, fungi, yeast, and bacteria synthesize the secreted IBPs, thanks to which they can live in the water channels created by these proteins (Figure 1C). Microorganisms at the subzero conditions in the water channels can respire, take up nutrients, multiply, and play an important role in biodegradation of organic compounds in polar regions [7]. Protection against the negative effects of frost through the production of IBPs is more widespread in bacteria. At least a dozen bacterial strains have been isolated that are able to synthesize proteins with TH, IRI, or ice-nucleating activity.

##### AFPs from Fungi

It has been demonstrated that only two yeast strains (*Glaciozyma* sp. AY30, *Glaciozyma antarctica*) and three snow mold fungi (*Typhula ishikariensis*, *Antarctomyces psychrotrophicus,* and *Coprinus psychromorbidus*) are able to synthesize these proteins at subzero temperatures (Table 1). The first discovered AFP (named LeIBP) of yeast origin came from the *Glaciozyma* sp. AY30 (formerly *Leucosporidium*) and was isolated from ice core sample of a freshwater pond near the Dasan station, Ny–Ålesund, Svalbard archipelago, Norway [23]. LeIBP is a glycosylated protein with molecular mass of 25 kDa that exhibits TH and IRI activity. The experimental model of the 3D structure of protein consists of a dimeric right-handed β-helix fold which is made of a large coiled structural domain, long helical region, and C-terminal hydrophobic loop [60]. In addition, the LeIBP gene was expressed in two expression systems: *E. coli* and *P. pastoris* in the pilot scale (700 L) and the obtained yields were at 300 mg/L [61]. Moreover, comparison of the activity of two forms of recombinant LeIBP, glycosylated and non–glycosylated, did not significantly affect the differences in TH and IRI activities. Glycosylation does not affect the structure and functionality of this protein. Another producer of AFP from psychrophilic yeast is *Glaciozyma antarctica* PI12. This strain produces two isoforms of antifreeze proteins: Afp1 with molecular mass of ~18 kDa [62] and Afp4 with molecular mass of ~25 kDa [63]. Afp1 shows 30% similarity of gene sequence with AFP from snow mold fungus *Typhula iskariensis* and Afp4 shows 93% similarity with LeIBP (*Glaciozyma* sp. AY30). Both Afp1 and Afp4 display the activity of TH and IRI. In 2018, Villarreall et al. demonstrated that three other species of Antarctic yeast (*Leucosporidium creatinivorum*, *Candida parapsilosis,* and *Goffeauzyma gastrica*) show antifreeze activity, but these proteins were not characterized [64].

*Typhula ishikariensis* produces seven secreted isoforms of AFPs abbreviated as TisAFPs. Crystal structures of two of them were solved and deposited in PDB: isoform TisAFP6 (PDB: 3VN3) [21] and a hyperactive isoform TisAFP8 (PBD: 5B5H) [65]. The molecular mass of TisAFPs is about 24 kDa [22,66]. The second fungal AFP from *Antarctomyces psychrotrophicus* (AnpAFP) is an extracellular protein of approximately 28 kDa [22]. AnpAFP exhibited two activities: TH (0.42 °C for 0.48 mM protein concentration with the maximum activity under alkaline conditions) and IRI (at 0.1 mg/mL protein concentration) [22].

##### AFPs from Bacteria

In recent years, many AFPs have been discovered from bacteria found in cold habitats (Table 1). Unfortunately, most of them have not been characterized yet. As in fungi, AFPs derived from bacteria have low TH activity with IRI activity. The first test relating to bacteria was published by Gilbert et al., 2004, in the experiment aimed at screening 866 isolates of bacteria from four different Arctic lakes. Only 19 species of bacteria that exhibited IRI activity were characterized [67]. The first discovered bacterial AFP was the protein from sea ice Gram-negative bacterium *Collwellia* strain SLW05 (ColAFP) [20]. In 2014, this protein has been characterized in biochemical and structural terms, displaying structural similarity to the structures of other AFPs originating from psychrophilic microorganisms (including LeIBP and TisAFP) [36]. Based on the sequential alignment of these proteins, it was found that eukaryotic organisms received the gene of these proteins from bacteria through horizontal gene transfer. In 2008, Garnham et al. published the first solved fragment structure of bacterial AFP isolated from Antarctic lake [68]. *Marinomonas primoryensis* produces a > 1 MDa protein with calcium-dependent activity (Figure 1E). The model of the 3D structure of this protein shows the bound water molecules associated with the binding site, which perfectly matches the network of ice crystals. The determination of this structure allowed explaining of the mechanism of ice binding by anchored clathrate waters [16,17], which are regularly arranged around the ice-binding surface (Figure 2J). Recently, more AFP-derived bacterial proteins have been discovered. Arctic bacteria *Flavobacterium frigoris* PS1 (FfIBP) synthesize the AFP, which has 56% structural similarity to AFP derived from *Leucosporidium* yeast (LeIBP), while having 10 times more TH activity than LeIBP [69]. Higher TH activity results from the structural differences of these proteins [70]. The FfIBP at the IBS has a characteristic motif (T–A/G–X–T/N) and more matched amino-acid residues to the binding site than LeIBP [70].

### 3.3. INPs

Unlike AFPs, INPs are large, hydrophilic, and multimeric proteins with subunits from 120 to 150 kDa in weight [76]. Most often, these proteins are associated with the cell membrane. Structurally, INPs are different from AFPs, but there is a speculation that they have a similar mechanism of action [77]. The best known and described INPs have been found on the outer membrane of several bacterial plant pathogens (*Pseudomonas syringae*) [78].

Based on the models of 3D structures of these proteins, INPs consist of three domains; the first—N–terminal, with a role anchoring the protein in the cell membrane, the second—the region of 16 amino-acid tandem repeats (CRD—central repeating domain), third domain—C–terminal, which function remains unknown [76]. These proteins differ significantly in the amino-acid composition and the repetitive sequence in the CRD region. The hydrophobic globular N–terminal domain covers only 15% of the INPs sequence. Its function is binding to lipids, polysaccharides, and other INPs. The interaction of the INP with the phospholipid membrane allows it to anchor in the cell membrane, aggregate, and organize the nucleation activity. CRD, as the most important part of INPs, is considered to be the place of interaction with ice. It consists of 3 domains of 16 amino acids. Although the entire INP is hydrophilic, the CRD fragments are flat and relatively hydrophobic. The last 15% of the protein is occupied by the C–terminal domain, the function of which is not yet known.

## 4. Mechanisms of Action of IBPs

### 4.1. Antifreeze Activities of AFPs

AFPs exhibit two activities that are related to their affinity to ice: TH and IRI (Figure 1B–D). In normal conditions (without the addition of AFPs), water melts and freezes at the same temperature which is equal to 0 °C. TH is the difference between melting and freezing points caused by micro curvature on the ice surface (Figure 1B–C). The water molecules are thermodynamically more difficult to join to a curved ice surface than to the flat one. This property is described by the Kelvin effect [79]. AFPs decrease the freezing temperature below the melting point in a non–colligative manner. The melting temperature also insignificantly rises from the melting point standard. AFP binds to the ice and causes the TH—this association causes ice to be protected against the melting as well as further freezing. TH is the sum of a large freezing hysteresis and a small melting hysteresis [80]. TH changes the morphology of ice crystals (Figure 1B). Ice crystals take the shapes ranging from circular, hexagonal, bipyramidal, to needle-shaped [18]. Value of TH is different between organisms [81]. Some insect AFPs have from 10 to 30 fold greater activity than fish AFPs at the same concentration [82]. The explanation of this phenomenon is still being established; probably IBPs of insect origin may bind to different planes of ice, not only to the basal one. Generally, TH values are in the range from 0.4 to 2.0 °C in fishes, from 3.0 to 6.0 °C in insects, from 0.15 to 0.7 °C in plants, and from 0.1 to 0.7 °C in microorganisms. Interestingly, some psychrophilic bacteria (*Pseudoaletoromonas*) and yeast (*Glaciozyma* sp. AY30) additionally produce enhancer molecules such as polysaccharides (e.g., trehalose), amino acids (e.g., glycine and betaine), or salts (e.g., NaCl), which increase the TH of AFPs [24,83,84].

Ice recrystallization can be described as the process of creation the larger ice crystals at the loss of smaller ones via the Kelvin effect [85]. The second activity of AFPs—the IRI activity (Figure 1D)—prevents the organelles and cells from the damage and death caused by large ice grains. The phenomenon of recrystallization typically occurs during defrosting, when the temperature temporarily decreases and is associated with dehydration the cells and structural damages. The IRI effect is already present in the concentration of AFPs of the order of micromolar instead of TH when the millimolar concentration of AFPs is needed. This activity seems to be a potential defense mechanism against freezing in many psychrophilic organisms. IRI activity was found in bacteria, plants, snow molds and yeast. In some plants, AFPs help to tolerate the freezing rather than protect against it. Thanks to this activity, the AFPs were examined for potential cryoprotective substances [86]. The IRI activity cannot be measured; it is only observed and recorded under the microscope and reported as an estimate. There are techniques used in evaluation of IRI activity, e.g., splat assay, sucrose sandwich splat assay, and capillary assay [87]. A large number of techniques were applied in order to discover how the AFPs influence the shape of ice crystals, e.g., ice etching, microscopy, fluorescence microscopy, cryometry. The first method was used to observe to which plane of ice (basal or prism) the IBPs bind. This technique revealed that fish AFPs are attached only to prism faces while insect AFPs bind to both prism and basal planes [88]. This indicates that these IBPs/AFPs have an affinity to specific planes of ice crystal depending on their origin. This fact could be explained on the molecular level by differences in primary structures of AFPs: mostly, composition of amino acids, their 3D arrangement, and hydration. The most important factor is the specific arrangement of especially threonines. Further studies [89,90,91] on this issue showed more than two planes of ice that IBPs may bind to, and these were basal, primary prism, secondary prism, and pyramidal prism. The plurality of these planes caused different values of TH activity measured by cryoscopy and sonocrystallization in six examined AFPs of various origins bound to the ice faces specific for them. It was showed that ice crystal growth is faster along its a-axis (primary prismatic plane) than along the c-axis (basal planes). That fact explains the bigger value of TH activity of AFPs that were bound to primary prismatic planes. It is also a reason for the growth of ice crystals in fish along the c-axis, in insects along a–axis, and in plants along both the c-axis and a-axis (Figure 1B) [91]. Despite these differences, AFPs’ TH and IRI activities require a region for unique protein–ice association that is a common feature and also identifier for AFPs. This region is called the ice binding site (IBS). In most of the AFPs, the IBS is relatively flat and hydrophobic. This feature is a common AFP identifier. Additional, IBPs must maintain rigidity during ice binding [29]. AFPs are soluble in aqueous solution at millimolar concentrations. In living organisms they use their property of lowering freezing point in a non-colligative manner, which means that their concentration does not affect that activity.

Most of AFPs have in their structure internal asparagine ladders, an external α-helix, or an extensive network of hydrogen bonds in the protein core, which keeps them rigid [21]. Another quite common feature among AFPs is the presence of some the repeating motifs in protein sequences. These repeats are non-specific but structurally similar, so that ice binding residues are aligned on one side to create an ice binding face. Moreover, for the purpose of adsorption, the distribution of polar and non–polar residues on the IBS must correspond to the atomic distribution of water molecules on the ice crystal surfaces [92]. Non-binding regions of proteins can be found outside, serving for the adverse interaction with water, to prevent it from being absorbed by the growing crystals adjacent to the water. AFP from the longhorn beetle (*Rhagium inquisitor*, RiAFP) was crystallized as a dimer which, however, was not stable in solution because of higher shape complementarity for RiAFP and ice than for two subunits of RiAFP’s crystallographic dimer [28].

According to the all of the above, three hypotheses of possible ice binding mechanisms of AFPs could be taken into consideration. The first hypothesis about IBPs binding to ice surfaces arose in 1977 based on research of DeVries and Raymond on fish AFPs type I and AFGP [93]. It was refined by study of anisotropic ice surface energy and the polymeric nature of IBPs and assumed that AFPs bind irreversibly following the ice surface by hydrogen bonds between the hydroxyl groups of the side chains of threonine residues in IBS [94]. DeVries’ hypothesis was levered by mutagenesis of two and four centrally placed threonines to serines, because AFPs’ activity lowered to 0–10% (in comparison to the wild type AFPs). On the other hand, when threonine residues were exchanged to valines, the loss of activity dropped to about 15% [95]. This fact brought the new hypothesis of the hydrophobic effect, which assumed that molecules of water located between protein and ice are pushed out to bulk solvent and AFPs bonding to ice is created because of high entropy. Other essential aspects were also examined, e.g., hydrogen bonding, IBS planarity, structural match with the ice-lattice, and inclusion of methyl groups within the ice lattice and ice-like waters. Considering these factors, the clathrate hypothesis was formed and assumed that water molecules form “cages” around hydrophobic region in IBS and then freeze as the next layer of ice on its surface, resulting in binding AFP and ice [96]. Recent fluorescent microscopy studies on Tm–AFP–GFP and AFP–GFP type III proved that AFPs binding to ice is not necessarily irreversible but semi–reversible [97]. Another two–step adsorption mechanism of one of them was the theory of surface–solution equilibrium. All of these mechanisms turned out to be untrue after the outcomes of research on heating of AFP [98]. Moreover, in 2016 Liu et al. reported an interesting mechanism (Janus effect) of both depression and promotion effects of AFPs on ice nucleation and binding to the ice [99]. The study investigated the effect of the ice-binding face (IBF) and non-ice-binding face (NIBF) of AFPs on ice nucleation via binding of AFPs to solid substrates in the way the sites are exposed to the liquid water. They concluded that the nucleation process is continued by IBF and the NIBF depressed the ice nucleation. This phenomenon is observed only in three representative AFPs: type III fish AFP, bacterial AFP (*Marinomonas primoryensis*), and a hyperactive insect AFP (beetle *Microdera punctipennsis dzungarica*) [99]. Molecular dynamics simulation analysis revealed that water molecules on the IBF form ice-like interfacial water structure due to the special arrangement of hydrophobic methyl and hydrophilic hydroxyl groups on the IBF. In contrast, almost no ice-like water structure is formed on the NIBF, which is possibly due to the absence of regular hydrophobic/hydrophilic patterns as well as the existence of charged groups and bulky hydrophobic groups. This work allows for a better understanding of the ice nucleation process involving AFPs [99].

It is also worth adding that in the case of AFGPs, the disaccharide residue (galactose N-acetylgalactosamine) plays an important role in binding to ice [100]. Meister et al. showed that the AFPG-borate complex is able to bind to ice, however, the adsorption rate is reduced by 65% compared to pure AFPG, with resulting decreased TH values. Sugar residues blocked by borate prove that glycosylation plays a more important role when AFPGs adsorb to ice rather than the hydrophobic groups associated with the peptide backbone.

### 4.2. Ice Nucleation Activity of INPs

In the group of IBPs, in addition to proteins that inhibit the growth of ice crystals, there are proteins that have the ability to initiate ice crystallization at temperatures from 0 to −2 °C [76]. INPs are the high molecular weight proteins for which the ice-binding site becomes a site for attaching free molecules of water, making them new ice crystal nuclei. Ice nucleation activity is widespread in many Gram-negative, pathogenic, and epiphytic bacteria. Interestingly, these bacteria can be psychrophiles (living at 0–15 °C) or mesophiles usually living at 30–37 °C but able to survive occasionally at lower temperatures by means of synthesis of e.g., cold-shock proteins (CSP) [76,101]. Numerous of INPs-producing bacteria have been reported, however, these proteins are not well characterized. INPs have been also reported in plants and insects and helped the organism survive freezing by controlling the location of ice [34]. They produced extracellular ice crystals and saved the cytosol of organisms from freezing. INPs act as nuclei of ice crystals by exhibiting a surface structure that is similar to the surface of ice crystals. They cause the formation of ice in the intercellular spaces, while protecting the internal water from freezing [34]. AFPs can inactivate INPs by interacting with them in the same way as with ice. There are speculations that these complexes more effectively inhibit the growth of already existing ice crystals.

The mechanisms that use INPs to create ice crystals are still not solved in contrast to the effect of IBPs on inhibiting the growth of ice crystals which is now well understood [76]. Numerous attempts have been made to express and purify these membrane proteins, however, due to their large sizes, they tend to misfold and aggregate. In addition, measuring the ability to nucleate ice is a complex process and depends on many factors.

## 5. Potential Applications of IBPs

### 5.1. AFPs

The discovery of AFPs resulted in an increasing interest in their applications, which are concentrated in the areas of food technology, agriculture, cryobiology, and material technology (Figure 3).

#### 5.1.1. Food Processing

One of the most important industries where AFPs can be applied is food manufacturing. AFPs used as a food supplement improve the quality of stored frozen products, e.g., by keeping a smooth texture without the ice grains of ice cream or by increasing the time of storage of meat, fish, fruits, dough, and vegetables. Many patents can be found that describe AFPs in food storage [102,103].

The most common problem in the production of ice cream is the ice recrystallization caused by temperature differences during product storage. Small ice crystals melt and turn into larger ones when the temperature decreases. Application of AFPs changes the texture and organoleptic characteristics of ice cream and improves them.

AFPs type III derived from psychrophilic fish were allowed for consumption in ice cream in 2008 by the European Food Safety Authority in Europe and in 2013 by the Food and Drug Administration in USA [103,104]. AFPs reduce the size of these crystals, so that the texture is better in the mouthfeel. Regand and Goff in 2006 were among the first who confirmed that theory by investigating IBPs—ice structuring proteins derived from cold-acclimated winter wheat grass—as an extract and named in short AWWE. IBPs in AWWE were tested in ice cream in the presence of stabilizers—skim milk powder, corn syrup solids, and locust bean gum. The size of ice crystals was observed using bright field microscopy. It was proven that IBPs demonstrate high activity when ice crystal growth was limited in the examined solutions. Just 0.13% total protein from AWWE concentration decreased the ice recrystallization rate by 44% for the most complex mix. Even smaller concentrations of proteins were needed for reducing this rate by 40% and 46% in the heat–shocked ice cream, namely 0.0025% and 0.0037% total protein from AWWE. It was observed that the simultaneous presence of the stabilizers and IBPs caused higher activity of proteins due to synergistic effects. An additional fact was that applying IBPs and heat-shock storage methods made the texture of ice cream smoother during sensory evaluation [104].

Zhang et al., 2016 used AFPs isolated from cold-acclimated oat (*Avena sativa*). The research comprised purification and identification of AFPs, called AsAFP, and analysis of TH activity. After the discovery that purification decreases TH activity, it was assumed that the reason was washing out low mass substances like activators, sugars, and antibodies that could enhance the positive effect of AFPs. IRI activity of AsAFP in ice cream was triggered by applying 0.1% (*w*/*w*) AsAFP or BSA to the samples, fast cooling and heating, fluctuating temperature, and observing the effects using a microscope. After violent cooling, there was no spectacular change in ice crystal size but during temperature fluctuation, the difference between the sample with AsAFP and with BSA was observed. There were smaller ice crystals in ice cream with addition of AsAFP than in ice cream with addition of BSA. Another measured parameter was melting ratio which is affected by many factors like ice cream composition, ice crystal size, consistency [105], and network of fat globules [106]. It was proven that applying 0.1% of AsAFP increases melting resistance by lowering ice crystal size, extending the time of the flow out of water from the thickened structure [107].

Kaleda et al., 2018 tested the impact of AFPs extracted from winter rye leaves (*Secale cereale*) of cultivar “Visello” on texture and microtexture of low-fat ice cream. The obtained extract contained 16.4% (*w*/*w*) total protein, soluble polysaccharides, and salts. The assay of AFP activity was also done using recombinant AFP with C-terminal residues YPPA mutated to YAAKDEL to improve protein solubility. The expression was performed in *E. coli* BL 21 strain. Two sets of samples were prepared—one of them was enriched with winter rye extract containing up to 400 mg/L lyophilized extract dry weight total protein and the second one was pricked with *E. coli* extract containing up to 35 mg/L IBPs. The ice cream was frozen to −5 °C then hardened to −40 °C for 24 h. After that samples were stored in −18 °C. Another study that was carried out on the ice cream utilized the measurement of IRI activity by using a modified sucrose sandwich assay method [104]. Temperature range was selected to maximize the ice recrystallization velocity and around melting temperature of sucrose solution that was used to make the ice cream. Occurrence of IRI activity was proven in both samples by curve inflection point measurement that corresponded to 50% IRI activity. For winter rye extract and recombinant AFP type III in *E. coli* it amounted to 0.17 mg/mL and 0.10 µg/mL, respectively. There is a hypothesis that recombinant AFP’s IRI activity is higher than observed by other groups because in this experiment ice cream mix was used. Microstructure of ice cream was observed by using polarized light microscopy and in comparison with the control sample—pure ice cream—ice crystals in ice cream with addition of IBPs were tinier as expected but also aggregated into small groups. The occurrence of these agglomerates would suggest that new structures from ice crystals formed by AFPs arise. In order to check the sweetness, roughness, and friability of the samples, a panel comprising seven trained sensory assessors was conducted. Assessors could check the mouthfeel only of the ice cream with winter rye extract and control group. Furthermore, the hardness of both sets of samples and control group was tested by applying the method proposed by Valera et al., 2014 [108], which relies on pinching samples with a special 3-mm diameter cylindrical probe at a speed of 2 mm/s to a depth of 15 mm and measurement of the force needed. The samples were stored at −18 °C and transferred to 21 °C for exactly 10 min. The concentrations between 3 and 25 mg/L of AFPs were used. Applying AFPs makes ice cream five times harder than the control group under a 3 mg/L concentration of protein and 13 times harder under a 25 mg/L concentration of protein. That fact was also confirmed by the sensory panel. The assessors claimed ice cream with addition of AFPs was more friable and rough than the control group. This could be explained by ice crystal aggregations formation [109].

The AFPs derived from fish are currently used in dairy products of Unilever, Häagen-Dazs, and Eskimo including ice cream and yogurt [110]. It may be convenient to apply AFPs in yogurt because of the possibility to produce recombinant AFPs by lactic acid bacteria naturally occurring in this product [111].

Because of the necessity of prolonging the shelf life of products, in the 1960s frozen dough was launched. Although frozen dough allows for extended storage and transport of the baked products further than the local bakeries, finished products, e.g., bread, lose the qualities such as uniform crumb, volume, and sensory qualities like odor, taste, and crust crispiness [112]. Forming ice crystals in frozen dough causes changes of salt concentration and pH that weaken yeast cells and may lead to their lysis. This in turn lowers pH of the product due to damage and leakage of the content of the yeast cells, which reduces fermentation capability and elongates the process. Moreover, destroyed yeast excrete glutathione that weakens disulfide bonds in glutenins which form gluten [113]. As a result, gluten depolymerizes which causes disintegration of cross-linked structure ensuring e.g., gas retention capacity. Another important consequence of freezing is water loss, which was prevented in many countries by application of potassium bromate until it was found to have a negative impact on consumers’ health.

The positive impact of AFPs on frozen dough was verified in 2007 [114,115]. The scientists used DcAFP derived from carrot (*Daucus carota*) which constituted 15.4% (*w*/*w*) of the added concentrated carrot protein (CCP). It was proven that in comparison with other cryoprotectants—BSA (bovine serum albumin) and SPI (soy protein isolate), samples with DcAFP after baking were characterized by a greater volume of loaf, deeper color of crust (as a result of the Maillard reaction induced by presence of AFPs), whiter crumb, and softness and smoothness in mouthfeel. Application of DcAFP lowered hardness of the dough because AFPs captured water molecules (even they were not frozen on the ice surface), improved fermentation capacity, and lead to forming less ice crystals in comparison with control samples by inducing TH activity. In addition, it was found that application of carrot juice lead to better taste and smell than using a solution of purified AFPs [88]. Similar results were received by using AFP derived from winter wheat [114] and barley [116].

It was also determined that DcAFP at a concentration 1.29 mg/mL on frozen white salted noodles enhanced the cooking properties and improved noodle texture [117]. Application of DcAFP protected the cross-linked structure of gluten from frost and temperature fluctuations, which was detected by observing the microstructure of noodles.

Not only AFPs were investigated to act as protectants during storage of frozen dough but also extracellular INPs at a concentration of 2.4 × 10^6^ units per gram of dough were considered. These IBPs were derived from *Erwinia herbicola* also known as *Enterobacter agglomerans* [113]. Unlike AFPs, the INPs protect the structure of frozen dough by aggregating free water molecules in ice crystals, preventing water escape after thawing and yeast cell damage (which is confirmed by high pH). The fermentation capacity reaches a high level and the volume of the bread loaf is bigger than that of the control sample.

The biggest problem during meat freezing is the formation of big, needle-shaped ice crystals that damage cell membranes, causing significant water loss. Assays of application of AFPs as cryoprotectants of frozen meat relied on IRI activity, which resulted in small ice grain formation, which prevented destruction of the cells. One of the preliminary ideas was to soak the samples of meat in the AFP solution before freezing. During this experiment, the samples were soaking too long and spoiled. In order to avoid similar situations, injection the AFP solution to living lamb before slaughter was suggested [118]. The concept was based on research from 1995 that considered intravenous injection of AFGP solution from Antarctic cod to living lambs. The animals were injected with 10 mL of examined solution to achieve concentrations of 0, 0.01, 1, 100, or 10,000 µg/kg liveweight [119]. After the slaughter the samples were packed in vacuum packages then stored at −20 °C for 2–16 weeks. Best results were accomplished by injection 24 h before slaughter at 0.01 µg/kg AFGP concentration. The lowest concentration required for effective action of AFP means that cost of frozen meat storage might significantly decline by AFP application [8]. Besides loss of water from frozen tissue, producers must face the problem of freeze-induced denaturation of the muscle proteins.

Recently used cryoprotectants, 4% sucrose or sorbitol, cause sweetness of meat, which is an adverse effect because of taste and medical reasons. That is another superiority of AFPs which lack flavor [120].

Until present there was no research on the impact of AFPs of different sources on freezing and storage of fresh fruit or vegetables. The only report was about carrot being preserved by synthetic peptides that have similar properties to IBPs [121]. Cubes of carrot with dimensions of 1 × 1 × 1 cm were soaked in solutions that contained three different peptides prior to freezing. Based on this research, it was found that application of these peptides lowers the drip losses, protects the color, structure, texture, and fugacity of frozen carrots in comparison to samples that did not contain these synthetic peptides. This research corroborated the validity of the use of food additives based on AFPs to prolong shelf life of frozen food and to improve its quality. Currently the only area where AFPs are used as food additives is ice cream sundae production in the USA, Australia, and New Zealand [122].

#### 5.1.2. Cryopreservation

The next potential and the most promising area of application of AFPs is cryopreservation. Transplantation and cryosurgery require new natural methods that facilitate the storage of tissues and organs at low temperatures. In order to extend the life of cells, or to protect them from damage caused by ice crystals, two types of chemical substances were used: cell penetrating cryoprotectants (ethylene glycol (EG), dimethyl sulfoxide (DMSO), and glycerol) and non–penetrating cryoprotectants (glucose, sucrose, trehalose, polivinylpyrrolidone, and polyvinyl alcohol). Recent studies have shown that AFPs can potentially replace these agents, because they are less toxic and non–penetrating. Moreover, they effectively inhibit the growth of ice crystals at lower concentrations than conventional substances; however, a deeper insight into the mechanism of freezing damage is required.

The first reports on the use of AFPs as cryoprotectants date from the 1990s when proteins from Antarctic and Arctic fish were investigated [123]. Since then, proteins derived from cold-blooded fish have been used in numerous studies involving cryopreservation [124]. This review is focused on research using AFPs of microbial origin, due to the high application potential of proteins of this source. Publications describing the AFPs from yeast or bacteria are still a minority of all publications using these proteins as cryoprotectants. The first report on the use of LeIBP (*Glaciozyma* AY30) appeared in 2011 and related to the positive effect of the addition of this protein on the osmotic resistance, motility, and viability of wild boar sperm during freezing [125]. Subsequent studies from 2012 used recombinant AFPs derived from *Glaciozyma* AY30 yeast on red blood cells [10]. The addition of LeIBP protein at a concentration of 0.4 mg/mL and 0.8 mg/mL in 40% glycerol significantly reduced the number of cells that have undergone haemolysis after thawing in rapid warming (45 °C) or slow warming (22 °C) compared to the control. Moreover, the addition of this protein as a cryoprotectant did not cause significant differences in morphological characteristics and cell size. These results indicate LeIBP’s ability to inhibit recrystallization of ice and to avoid critical electrolyte concentrations that lead to cell damage. Subsequent reports from 2015 also use the recombinant AFPs derived from the psychrophilic yeast *Glaciozyma* sp. AY30. Koh and co–authors used the marine diatom *Phaeodactylum tricornutum*, which is a model organism and has already been the object of many ecological, physiological, biochemical, and molecular studies [60]. When cells were frozen using a two–step method, cell survival after thawing on day 11 was significantly increased in the presence of 0.1 mg/mL LeIBP in 10% polyethylenic glycol (PEG) and 10% EG. Furthermore, the authors reported that the in vivo concentration of chlorophyll substantially increased after the addition of 0.1 mg/mL LeIBP to 10% DMSO, PEG, and EG. Additional studies using scanning electron microscopy showed that the cells were effectively preserved and the epitheca or hypotheca was not deformed. Kim and co-authors showed that the addition of recombinant LeIBP can significantly improve the viability of cryopreserved mammalian cells by at least 10% [125]. Moreover, the addition of AFP reduced the concentration of toxic compounds such as DMSO below 5%. Also in 2015, Lee and co–authors published a report where they compared the impact of three different proteins: LeIBP (*Glaciozyma* sp. AY30), FfIBP (*Flavobacterium frigoris*), and type III AFP (*Zoarces americanus*) on cryopreservation of ovaries frozen by vitrification, and their subsequent warming and transplantation into mice. The results showed that supplementing AFPs at a concentration of 20 mg/mL in the vitrification solution had a protective effect on the survival of ovarian tissue during cryopreservation and transplantation, and that the most beneficial activity was observed for the LeIBP [11].

An important application of AFPs in cryopreservation is also hypothermic organ storage [12]. However, AFPs of animal origin were used for these studies. A large scale application and clinical tests of AFPs require many laboratory and pilot scale studies. In addition, the cost of production and allergy problems should be taken into consideration. For now, AFPs derived from fish are the most studied and applied proteins, i.a., because of low risk of allergy. In summary, cryoprotection has become the most promising area of commercial application of AFPs.

#### 5.1.3. Agriculture

Another important branch of industry in which AFPs can be used is agriculture. So far, AFPs produced by plants resistant to temperature fluctuations (survive during extracellular freezing) have been characterized. Plants produce their own AFPs, however, with low TH and IRI activity, which do not completely protect them from the formation of ice crystals, contrary to AFPs in fish or insects. Attempts were made to express these proteins in plants to immunize them for periodic temperature decrease even to −5 °C. First reports on the creation of cold resistant plants come from the 1990s. A trial of AFP type I protein expression from marine fish was made using potatoes, tomatoes, and tobacco [126,127,128]; however, no satisfactory results were obtained. Over the years, many transgenic plants have been made to raise the efficiency of the expression of fish AFPs in plants. The maximum level of TH activity obtained was 1 °C more compared to the wild type. Due to the problems, the attention was directed to proteins derived from insects, which are characterized by much higher TH activity (even up to 12 °C). Holmberg et al., 2001 created the first, transgenic tobacco that expressed the AFP gene derived from brown spruce *Choristoneura fumiferana* [129]. The transformation was successful to achieve TH of 0.37 °C in apoplastic fluids and the IRI activity of IRI in leaf homogenate. During the 2000s, many attempts were made to increase the resistance of plants to low temperatures. In 2010, Zhang et al. created the *A. thaliana* with perennial ryegrass cv. Caddyshack (*Lolium perenne*) gene (LpIRIa, LpIRIb) which can survive the temperatures between 4 and −8 °C [130]. Another team in 2016 (Breadow et al.) achieved *A. thaliana* resistance to temperatures from −5 to −8 °C, also using a gene from perennial ryegrass v. Pacific Seed Diploid (*Lolium perenne*) [131]. The obtained results were satisfactory, as the controlled expression of AFPs allowed reduction of cell dehydration and intracellular membrane damage caused by freezing. Achieving higher freezing resistance of plants is a great challenge for researchers and must be continued due to climate change [132]. Unpredictable frosts in spring and autumn, due to climate change, create problems for farmers. Expression of AFPs can help protect plants that are sensitive to low temperatures, e.g., in sectors such as delicate ornamental crops, or plants with soft fruit. The prospects for the use of these unique proteins are extremely significant, so it is important that they consciously contribute to food safety and prevent global hunger by protecting sensitive crops.

#### 5.1.4. Other Applications

Except for the food industry, agriculture and cryopreservation, the areas of potential application of IBPs are still growing, e.g., in sectors of industry such as: material technology, the fuel industry, and weather control.

The ability of AFPs to change the morphology of ice crystals brought the idea of introduction of these proteins to a wide range of materials, including macroporous ceramic materials, polymers, and complex composites for the production of catalysts, absorbents, tissue engineering scaffolds, transport systems, and strong materials of low weight [30]. One example where the production of scaffolds and porous materials may be used is the transport of drugs and nutrients through controlled unidirectional freezes with subsequent lyophilization.

The next interesting use of AFPs is applying them as potential anti-icing substances [133]. Gwak et al., 2015 have developed a method for coating industrial metals (e.g., Al) with AFPs derived from cold-sea diatoms, *Chaetoceros neogracile* (Cn-AFP), to prevent the formation of ice crystals on refrigeration surfaces. They created a recombinant fusion protein consisting of AFP (Cn-AFP) and Al–binding peptides. After 3 h of freezing, no ice crystals were observed on Al bars coated with AFPs, compared to controls where the frosting was significant. The results of this experience can be used in kinds of metals in aviation or in refrigeration, where the formation of large ice crystals is a crucial parameter.

Finally, Wilkins et al., 2019 have recently demonstrated that AFPs can be conjugated onto polymer-stabilized gold nanoparticles. This may lead to the application of AFPs organized into complex assemblies in colloid science or in basic sciences utilizing the nanoparticle cores for tracking and bioimaging [134]. Moreover, researchers seek for a development of compounds that mimic AFP activity, for example small molecules, polymeric analogues (e.g., polyvinyl alcohol derivatives), or inexpensive metallic salts such as zirconium (IV) acetate [135]. Such materials could increase the commercial availability of antifreeze products, as AFPs are prone to denaturation and expensive to produce.

### 5.2. INPs

In contrast, INPs are widely used as factors affecting the process of atmospheric glaciation, as a consequence of precipitation and cloud formation. Currently, these proteins are used for the production of artificial snow. There is a commercial product called Snomax^®^, created from lyophilized, non-viable bacteria from *Pseudomonas syringae* species [136]. Protein extracts from *P. syringae* work as snow inducers and improve the crystallization process [137]. Apart from commercial applications, these proteins have been applied in research areas. INPs are used as fusion proteins because they have the ability to anchor in the cell membrane so that the expressed proteins are able to be exposed on the cell surface. However, the problem of misfolding or aggregation causes that only a few INPs can be used as biosensors or biosorbents [138]. Other studies include the use of INPs as reporter genes [139]. The level of transcription of the tested fusion protein is measured on the basis of the ice-nucleating activity of this protein. This method is based on a physical phenomenon compared to existing methods which are based on enzymatic reactions. The speed, simplicity, and high sensitivity of this method become extremely useful tools in various studies evaluating gene expression by microorganisms living in natural environments [140,141].

## 6. Future Outlook

Psychrophiles represent a significant group of organisms; hence they are an interesting object to study in academia and biomedical fields. Much is said about extremophiles in the context of the source of unique biomolecules used in different types of industry. Especially valuable from a technological point of view are the enzymes which due to their numerous structural and kinetics adaptations have already found many applications in food technology, molecular biology, medicine, etc. Furthermore, the new intelligent biopolymers, antibiotics, anti-cancer drugs, and IBPs isolated from cold-loving microorganisms also deserve attention. IBPs recognized as cryoprotectants are becoming increasingly desirable for the agri-food industry, material science, coatings, and biomedical applications. Their potential is certainly much greater, but not yet fully understood, despite the clear progress observed in the last years in the structural and functional characteristics of these biomolecules. However, there are still many unanswered questions related to e.g., the relations between IBP types, their activity, and ice binding mechanisms. Great hopes associated with this are found in the intensive development of genetic engineering, synthetic biology, and bioinformatics. The ultimate goal should be the manufacture of IBPs in high yields at low cost for scientific purposes and successful commercialization. It should be also remembered that control over the nucleation and growth of ice crystals is crucial for the survival of various species, atmospheric glaciation processes, and decisive for the structural integrity and properties of a broad range of water-based materials.

## Figures and Tables

**Figure 1 biomolecules-10-00274-f001:**
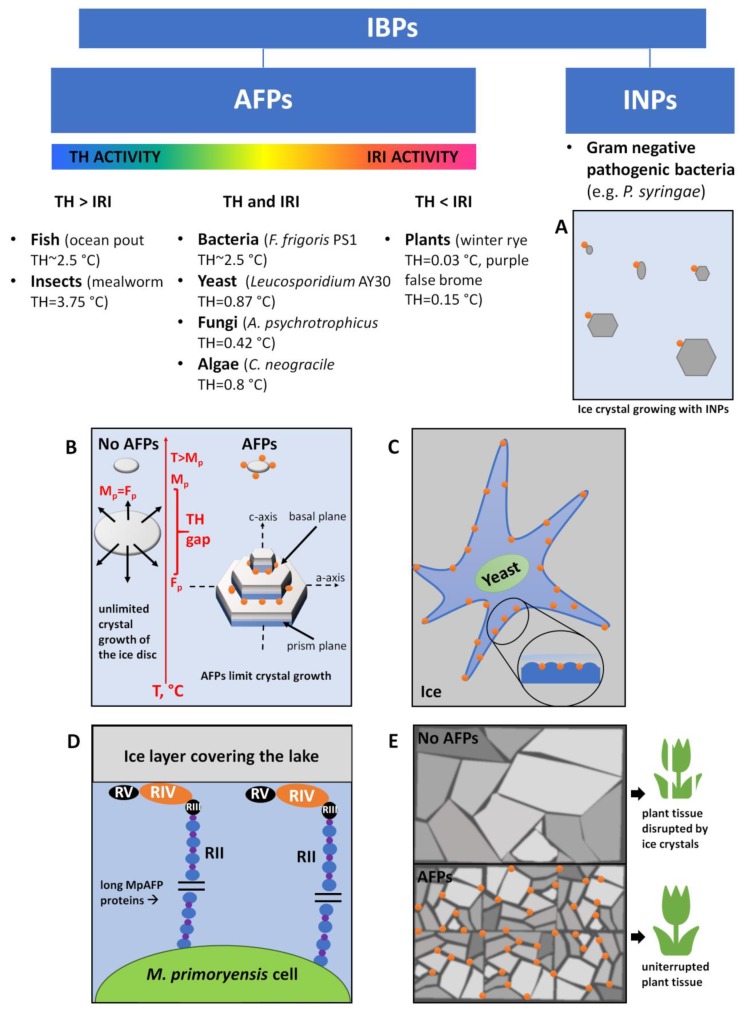
Classification, activities, and sources of the ice binding proteins (IBPs). IBPs can be divided into: (**A**) ice-nucleating proteins (INPs) that initiate the formation of ice crystals at high subzero temperatures, and (**B**–**E**) antifreeze proteins (AFPs) that decrease the freezing point of body fluids in organisms to avoid freezing. AFPs display the activities of: (**B**) thermal hysteresis (TH; M_p_ stands for melting point, F_p_—for freezing point), and (**E**) ice recrystallization inhibition (IRI); with some organisms displaying high TH, but low/no IRI activity (fish, insects), other—moderate TH and IRI activities (microorganisms), and plants exhibiting high IRI/low TH activity. Moreover, microorganisms produce secreted AFPs (**C**) that allow them to survive inside the ice crystal, and AFPs (**D**) that may keep the bacterium in water under the ice, assisting in providing an access to O_2_ and nutrients. IBPs are visualized by orange dots in panels (**A**–**E**), water is colored blue, and ice gray.

**Figure 2 biomolecules-10-00274-f002:**
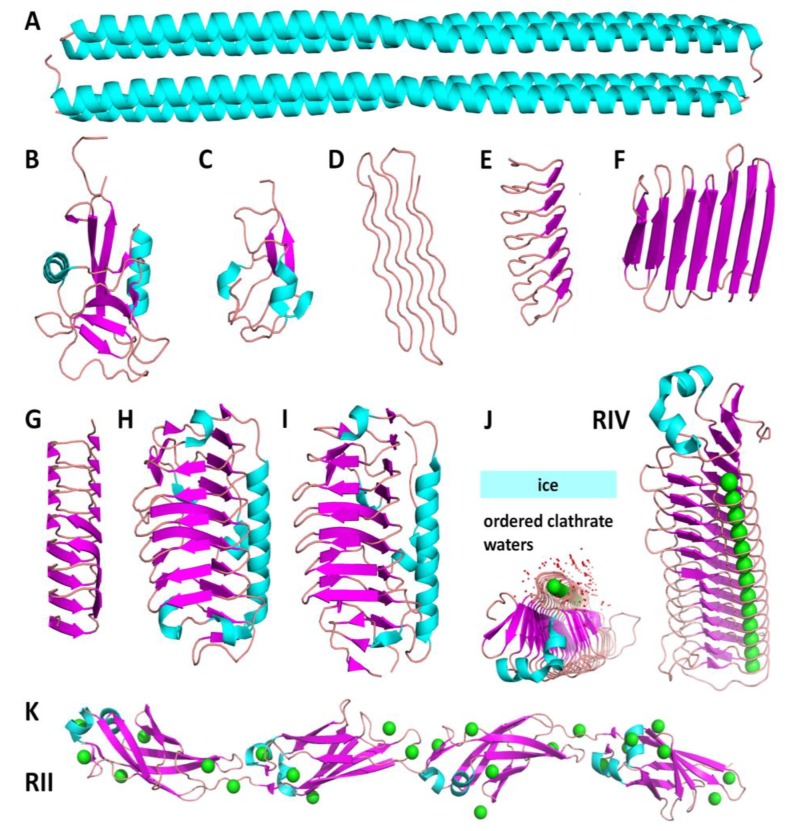
Structures of the antifreeze proteins (AFPs) of different origins. Models of the experimental structures were accessed from the Protein Data Bank (PDB) and presented as a cartoon view. Cyan indicates an α-helix, magenta—a β-strand, pink—a loop, red—waters, green—Ca^2+^ ions. (**A**–**C**) fish AFPs (**A**—hyper active type I AFP from winter flounder, PDB: 4KE2; **B**—type II AFP from longsnout poacher, PDB: 2ZIB; **C**—type III AFP from European eelpout, PDB: 4UR4). (**D**–**F**) insect AFPs (**D**—snow flea AFP, PDB: 2PNE; **E**—AFP from *Tenebrio molitor* beetle, PDB: 1EZG; **F**—AFP from *Rhagium inquisitor*, PDB: 4DT5). (**G**) Plant AFP from perennial ryegrass, PDB: 3ULT. (**H**–**J**) microbial AFPs (**H**—AFP from an Antarctic sea ice bacterium *Colwellia*, PDB: 3WP9; **I**—AFP from Arctic yeast *Leucosporidium*, PDB: 3UYV; **J**–**K**—MpAFP from *Marinomonas primoryensis*: RIV—ice binding with Ca^2+^ ions in a regular arrangement and ordered waters, PDB: 3P4G, and linker RII, PDB: 4P99).

**Figure 3 biomolecules-10-00274-f003:**
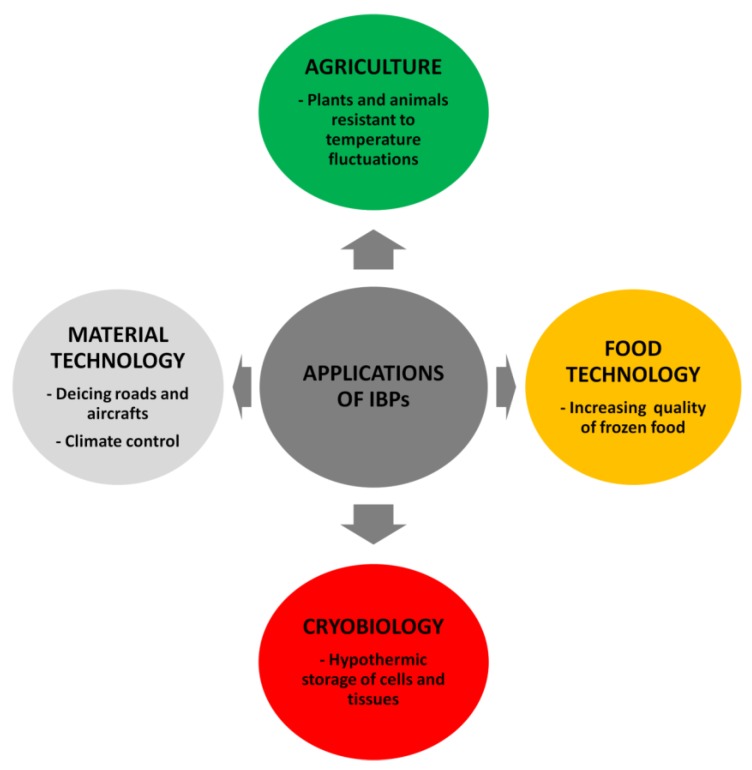
Applications of the ice binding proteins (IBPs) in different branches of industry.

**Table 1 biomolecules-10-00274-t001:** Examples of psychrophilic organisms and antifreeze proteins (AFPs) produced by them.

Organism (Protein)	Molecular Mass (kDa)	TH/IRI Activity	Gene/Protein Accession Nr	Place of Isolation	Ref.
**YEAST**					
*Glaciozyma* sp. AY30 (LeIBP)	26.80	+/+	ACU30807.1/C7F6X3	a freshwater pond, Ny–Ålesund, Svalbard archipelago, Norway	[23]
*Glaciozyma antarctica* PI12 (Afp1)	18.24	+/+	ACX31168.1/D0EKL2	sea ice, Casey Research Station, Antarctica	[62]
*Rhodotorula glacialis*	NA *	+/+	NA/NA	soils, mosses and algal mats, Great Wall Station, King George Island, South Shetland Islands; Zhongshan Station, Larsemann Hills, Prydz Bay and Soya coast, East Antarctica	[71]
**FUNGI**					
*Typhula ishikariensis* (TisAFP)	24.09	+/+	BAD02893.1/Q76CE6	Finnmark (northern Norway), Svalbard, Iceland, western Greenland, Siberia	[66]
*Antarctomyces psychrotrophicus*	28.00	+/+	NA/NA	mosses, soils and algal mats, Great Wall station, King George Island, South Shetland Islands; Zhongshan Station, Larsemann Hills, Prydz Bay, East Antarctica	[22]
*Coprinus psychromorbidus*	23.00	+/+	NA/NA	Finnmark (northern Norway), Svalbard, Iceland, western Greenland, and Siberia	[66]
**BACTERIA**					
*Collwellia* SLW05 (ColAFP)	26.35	+/+	ABH08428.1/A5XB26	winter sea ice, the west side of the Antarctic Peninsula	[20]
*Marinomonas primoryensis* (MpAFP)	1500	+/+	ABL74378.1/A1YIY3	Antarctic Lakes, Vestfold Hills, East Antarctica	[16]
*Flavobacterium frigoris* PS1 (FfIBP)	25.46	+/+	AFK13196.1/H7FWB6	sea ice, shore of McMurdo Sound, Antarctica	[69]
**PLANTS**					
ryegrass *Lolium perenne*	13.30	+/+	ACG63783.1/B5T007	native to Europe, temperate Asia, and North Africa; widely distributed throughout the world, including North and South America, Europe, New Zealand, and Australia.	[31]
wild carrot (*Daucus carota*)	36.80	+/+	AAV66074.1/Q5RLY3	Europe, southwest Asia, North America and Australia	[72]
**INSECTS**					
beetle (mealworm) *Tenebrio molitor*	8.40	+/+	AF160494.1/O16119	areas associated with human activity	[73]
longhorn beetle *Rhagium inquisitor*	12.54	+/+	HQ540314.1/E5LR38	Holarctic	[28]
snow flea *Hypogastrura harveyi*	6.50	+/+	ABB03725.1/Q38PT6	North America	[35]
**FISH**					
winter flounder *Pseudopleuronectes americanus*	19.34	+/+	ABX38716.1/B1P0S1	waters of the western north Atlantic coast, from Labrador, Canada to Georgia, United States of America	[40]
longsnout poacher *Brachyopsis segaliensis*	18.02	+/+	BAF37106.1/A0ZT93	Northwest Pacific Ocean	[74]
European eelpout *Zoarces viviparus*	6.90	+/+	ABN42205.1/A3EYI7	Northeast Atlantic; Baltic, Barents, Irish, North, and White Seas	[75]

* NA no data available.

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
