# Peer review of "Ice Binding Proteins: Diverse Biological Roles and Applications in Different Types of Industry"

_biomolecules, 2020, doi:10.3390/biom10020274_

Round 1

Reviewer 1 Report

Questions:

87: In "Some Arctic bacteria (Marinomonas primoryensis) maintain access to oxygen and nutrients by binding to the ice surface through ice–binding domains of large Ca2+–dependent adhesion proteins (MpAFP, 1.5 90 MDa) [7]." it is unclear to me how the ice binding of these proteins leads to an access to oxygen and nutrients.

General remarks:

1. To maintain a logical build-up of the review, I would suggest that the structure-describing part of section 4.2 will be transferred to a new section 3.3, and that in the title of section 3 the word "AFP" is changes with "IBP".

2. A table of contents would create a better overview of what is covered in the review.

3. On line 340 the authors state: "Most of INPs–producing bacteria have been reported". I am no expert in that particular field, but it seems impossible to me that one can know how many INPs–producing bacteria there are, and thus whether most are described in the literature or not. If so, it would be good to add a reference to the statement.

Some textual/graphics remarks:

35: "The definition of cold–adapted microorganisms includes two groups" => "Cold–adapted microorganisms can be divided into two groups" (I don't think a definition can include groups)

39: "persist" => live (persist could also be interpreted as "staying behind after the temperature change")

41: To what fact does "In addition" refer to? To the list of molecular-level changes listed at the end of the previous paragraph? Please make explicit (or remove "In addition").

62: "which" => "whose"

63: add hyphen in between "ice" and "nucleating", and in all other similar instances, e.g. "amino acid sequence" on line 115 should be "amino-acid sequence"  (see Rule 1 here: https://www.grammarbook.com/punctuation/hyphens.asp )

65: "bases" => "is based" (or another synonym as 'based on' is already used in the previous sentence)

71: "merge" => "merging"

76: "the influence of environment" => "environment"? (As I don't know what "the influence" adds here. If you do want to keep "the influence, add a "the" before "environment")

79: "to" => "into". Also explain M_p and F_p abbreviations in caption. What should the round gray discs below "No AFPs" and "AFPs" indicate? It'd make more sense to me if the latter disc was smaller and surrounded by a number of AFPs, thereby making them a zoomed out version of the ice crystals below them. And I suppose the blue thing next to the yeast cell is a side view? Maybe also make explicit (e.g./also using an icon like this one: https://www.netclipart.com/isee/ibRbxw_clipart-eye-side-view-eye-view-icon-png/ )

84: remove full stop at end of subsection title

94: "gain bacteria the" => "give bacteria"

103: a "bind" is not a thing, so another word/sentence is required (e.g. "they all bind irreversibly...")

106: "bundle"=> "bundles". Also on this line: I don't see a single alpha helix in Fig 2A, while this is suggested here.

107: add "the" in between "in" and  "Protein"

110: "The evolutionary pressure to live in harsh conditions caused that similar AFPs was " => "The evolutionary pressure of living in harsh conditions resulted in the fact that similar AFPs were "

112: "which was the adaptation to live" => "of adaptation to living"

113: "of AFP" => "of the AFP"

116: "some fish glycoprotein was" +> is it a single glycoprotein (then 'some' => 'a') or multiple (then "was" => "were")

More subjective preferences for the author's consideration:

20: "broad" => "large, polyfunctional"

30: "influence" => "effects"

53: "forming" / "damage" => "formation of" / "damages"

83: "at" => "in"

102: "freeze" => "freezing"?

105: "utterly" => "very" or "totally" would be more typical words in this context

341: "due to the structural similarity" => "by exhibiting a surface structure that is similar"

Author Response

Dear Reviewer,

We would like to gladly thank you for a very good overall score and your beneficial advices. We agree with the suggestions and believe that the changes introduced after your review will undoubtedly increace the quality of the manuscript. Please find our detailed responses to the particular comments below.

Comments and Suggestions for Authors

Questions:

87: In "Some Arctic bacteria (Marinomonas primoryensis) maintain access to oxygen and nutrients by binding to the ice surface through ice–binding domains of large Ca2+–dependent adhesion proteins (MpAFP, 1.5 90 MDa) [7]." it is unclear to me how the ice binding of these proteins leads to an access to oxygen and nutrients.

-> Thank you this comment, indeed we realised that an explanation was lacking. We added the sentences clarifying the matter after the sentence you cited:

„MpAFP is an extracellular, multidomain (Repeat I-V, RI – RV), “train-like” protein that at one end anchors to bacterium and at the other (by RIV) binds to ice (Figure 1D). Moreover, the long RII provides a distance between the cell and the surface of ice layer and prevents bacterium’s incorporation into it. Hence, the expression of MpAFP helps the strictly aerobic M. primoryensis remain in the upper reaches of the ice-covered lake where oxygen, nutrients and light are most abundant [17,18].”

We also added a panel to the Figure 1 (1D) and to the Figure 2 (2J-K), crystal srtuctures of MpAFP) and cited two references:

[17] Garnham, C.P.; Campbell, R.L.; Davies, P.L. Anchored clathrate waters bind antifreeze proteins to ice. Proc.Natl.Acad.Sci.USA 2011, 108, 7363-7367.

[18] Vance, T.D.R.; Olijve, L.L.C.; Campbell, R.L.; Voets, I.K.; Davies, P.L.; Guo, S. Ca2+-stabilized adhesin helps an Antarctic bacterium reach out and bind ice. Bioscience reports 2014, 34, 357-368.

General remarks:

To maintain a logical build-up of the review, I would suggest that the structure-describing part of section 4.2 will be transferred to a new section 3.3, and that in the title of section 3 the word "AFP" is changes with "IBP".

-> We introduced the suggested change. We believe this indeed helps to create a more logical structure. The part describing the structures of INPs is now in Paragraph 3.3.

A table of contents would create a better overview of what is covered in the review.

-> A table of contents has been added.

On line 340 the authors state: "Most of INPs–producing bacteria have been reported". I am no expert in that particular field, but it seems impossible to me that one can know how many INPs–producing bacteria there are, and thus whether most are described in the literature or not. If so, it would be good to add a reference to the statement.

-> Thank you for this comment, we removed the imprecise word “Most” and cited an article in which the authors list the described genera and species of bacteria which synthesize the INPs:

[101] Lorv, J.S.H.; Rose, D.R.; Glick B.R. Bacterial Ice Crystal Controlling Proteins. Scientifica 2014, 20, 976895. doi: 10.1155/2014/976895.

Some textual/graphics remarks:

35: "The definition of cold–adapted microorganisms includes two groups" => "Cold–adapted microorganisms can be divided into two groups" (I don't think a definition can include groups)

-> The sentence was rephrased.

39: "persist" => live (persist could also be interpreted as "staying behind after the temperature change")

-> The word was changed.

41: To what fact does "In addition" refer to? To the list of molecular-level changes listed at the end of the previous paragraph? Please make explicit (or remove "In addition").

-> „In addition” was removed.

62: "which" => "whose"

-> The word was changed.

63: add hyphen in between "ice" and "nucleating", and in all other similar instances, e.g. "amino acid sequence" on line 115 should be "amino-acid sequence"  (see Rule 1 here: https://www.grammarbook.com/punctuation/hyphens.asp)

-> The hyphens were added to all instances. Thank you for this precious language advice.

65: "bases" => "is based" (or another synonym as 'based on' is already used in the previous sentence)

-> The sentence was rephrased.

71: "merge" => "merging"

-> The word was changed.

76: "the influence of environment" => "environment"? (As I don't know what "the influence" adds here. If you do want to keep "the influence, add a "the" before "environment")

-> The phrase was changed to „environment”.

79: "to" => "into". Also explain M_p and F_p abbreviations in caption.

-> The word was changed. The abbreviations were explained.

What should the round gray discs below "No AFPs" and "AFPs" indicate? It'd make more sense to me if the latter disc was smaller and surrounded by a number of AFPs, thereby making them a zoomed out version of the ice crystals below them. And I suppose the blue thing next to the yeast cell is a side view? Maybe also make explicit (e.g./also using an icon like this one: https://www.netclipart.com/isee/ibRbxw_clipart-eye-side-view-eye-view-icon-png/)

-> The gray discs indicate the small ice crystals just above the Mp; we have clarified this at the Figure 1B and added AFPs surrounding the discs – indeed we find it more clear now too. We have also modified the panel with the yeast cell (Figure 1C).

84: remove full stop at end of subsection title

-> The full stop was removed.

94: "gain bacteria the" => "give bacteria"

-> The word was changed.

103: a "bind" is not a thing, so another word/sentence is required (e.g. "they all bind irreversibly...")

-> The sentence was rephrased.

106: "bundle"=> "bundles".

-> The word was changed.

Also on this line: I don't see a single alpha helix in Fig 2A, while this is suggested here.

-> The statement was changed.

107: add "the" in between "in" and  "Protein"

-> „The” was added.

110: "The evolutionary pressure to live in harsh conditions caused that similar AFPs was " => "The evolutionary pressure of living in harsh conditions resulted in the fact that similar AFPs were "

-> The sentence was rephrased.

112: "which was the adaptation to live" => "of adaptation to living"

-> The sentence was rephrased.

113: "of AFP" => "of the AFP"

-> „The” was added.

116: "some fish glycoprotein was" +> is it a single glycoprotein (then 'some' => 'a') or multiple (then "was" => "were")

-> The sentence was rephrased.

More subjective preferences for the author's consideration:

20: "broad" => "large, polyfunctional"

-> The word was changed.

30: "influence" => "effects"

-> The word was changed.

53: "forming" / "damage" => "formation of" / "damages"

-> The sentence was rephrased.

83: "at" => "in"

-> The word was changed.

102: "freeze" => "freezing"?

-> The word was changed.

105: "utterly" => "very" or "totally" would be more typical words in this context

-> The word was changed.

341: "due to the structural similarity" => "by exhibiting a surface structure that is similar"

-> The sentence was rephrased.

The cover letter is also attached in the online submission system.

Reviewer 2 Report

Ice binding proteins (IBPs), which generally includes antifreeze proteins and ice nucleating proteins, are a diverse class of proteins that show specific affinity to ice surfaces. They can protect the living organisms in subzero habitats from possible freezing injuries by inhibiting ice formation or recrystallization. IBPs have attracted the attention of researchers as a fascinating nature designed agent that regulating ice formation with high efficiency. The applications of IBPs in the food industry, in cryopreservation, and in other technologies are vast. The present review attempts to summarize the current state of art and possible utilization of the IBPs in different type of industries, which are interesting to scientists and engineers in ice science. However, the clarity of scope and the aim of the article are not clear enough. Detailed suggestions are as follows:

1)     The title of the review, which includes the words ‘innovative’ and ‘cryo-protectants’ is not accurate. It has almost been half a century since the first finding and application in cryopreservation of IBPs, thus ‘innovative’ is no longer the proper word to describe it. The word ‘cryo-protectants’ should not be included in the title since the wide application of IBPs reviewed in the present M.S. includes not only cryopreservation but also many other areas such as the food industry and the production of artificial snow.

2)     A review should include interpretation of recent advances. However, the present version only includes 2 papers in 2019, 1 paper in 2018 and 4 papers in 2017, and many other cited articles are researches about 10 or 20 years ago. More articles in recent 5 years should be reviewed so as to follow the latest advantages.

3)     In the abstract or introduction part, the author should state clearly the aim of the article. For example, if the review is to summarize the application of IBPs in different industries, the introduction part should be focused on this topic, rather than describing the cold adaptation of organisms in details which almost takes two paragraphs.

4)      Actually there are quite a few very nice reviews about IBPs in recent years, accounting for their structure-function relationships, action mechanisms, or application in cryopreservation. The present M.S. should address its specific focus which is different from the other reviews, and guide to other reviews if necessary.

(1)     Many relevant papers that shape the understanding of IBPs action mechanisms are missed in the present M.S., indicating a lack of comprehensive research of the author to this area. The following papers are some works that should not be omitted, e.g., Proceedings of the National Academy of Sciences 2016, 113 (51), 14739-14744; J Am Chem Soc 2018, 140 (30), 9365-9368; Proceedings of the National Academy of Sciences of America 2016, 113 (14), 3740-3745;Proceedings of the National Academy of Sciences of the United States of America 2016, 113 (14), 3714-3716.

Author Response

Dear Reviewer,

We would like to gladly thank you for a good overall score and your beneficial advices. We agree with the suggestions and believe that the changes introduced after your review will undoubtedly increase the quality of the manuscript. Please find our detailed responses to the particular comments below.

Comments and Suggestions for Authors

Ice binding proteins (IBPs), which generally includes antifreeze proteins and ice nucleating proteins, are a diverse class of proteins that show specific affinity to ice surfaces. They can protect the living organisms in subzero habitats from possible freezing injuries by inhibiting ice formation or recrystallization. IBPs have attracted the attention of researchers as a fascinating nature designed agent that regulating ice formation with high efficiency. The applications of IBPs in the food industry, in cryopreservation, and in other technologies are vast. The present review attempts to summarize the current state of art and possible utilization of the IBPs in different type of industries, which are interesting to scientists and engineers in ice science. However, the clarity of scope and the aim of the article are not clear enough. Detailed suggestions are as follows:

1)     The title of the review, which includes the words ‘innovative’ and ‘cryo-protectants’ is not accurate. It has almost been half a century since the first finding and application in cryopreservation of IBPs, thus ‘innovative’ is no longer the proper word to describe it. The word ‘cryo-protectants’ should not be included in the title since the wide application of IBPs reviewed in the present M.S. includes not only cryopreservation but also many other areas such as the food industry and the production of artificial snow.

-> Thank you for this advice. We changed the title to more suitable one: Ice Binding Proteins: Diverse Biological Roles and Applications in Different Types of Industry.

2)     A review should include interpretation of recent advances. However, the present version only includes 2 papers in 2019, 1 paper in 2018 and 4 papers in 2017, and many other cited articles are researches about 10 or 20 years ago. More articles in recent 5 years should be reviewed so as to follow the latest advantages.

-> We reviewed and added 9 recent articles (from the last 5 years) - references [1, 3, 89, 90, 99, 100, 112, 134, 135]:

Drake, H.; Ivarsson, M.; Bengtson, S.; Heim, C.; Siljeström, S.; Whitehouse, M.J.; Broman, C.; Belivanova, V.; Åström, M.E. Anaerobic consortia of fungi and sulfate reducing bacteria in deep granite fractures. Nat Commun 2017, 8, 55. doi:10.1038/s41467-017-00094-6 Fröhlich-Nowoisky, J.; Kampf, C.J.; Weber, B.; Huffman, A.; Pöhlker, C.; Andreae, M.O.; Lang-Yona, N.; Burrows, S.M.; Gunthe, S.S.; Elbert, W.; Su, H.; Hoor, P.; Thines, E.; Hoffmann, T.; Després, V.R.; Pösch, U. Bioaerosols in the Earth system: Climate, health, and ecosystem interactions. Atmospheric Research 2016, 182, 346-376. doi:10.1016/j.atmosres.2016.07.018 Olijve, L.L.C.; Meister, K.; DeVries, A.L.; Duman, J.G.; Guo, S.; Bakker, H.J.; Voets, I.K.. Blocking rapid ice crystal growth through nonbasal plane adsorption of antifreeze proteins, Proc Natl Acad Sci USA 2016, 113, 3740-3745. doi:10.1073/pnas.1524109113 Haji-Akbari, A. Rating antifreeze proteins: Not a breeze. Proc Natl Acad Sci USA 2016, 113, 3714-3716. doi:10.1073/pnas.1602196113 Liu, K.; Wang, C.; Ma, J.; Shi, G.; Yao, X.; Fang, H.; Song, Y.; Wang, J. Janus effect of antifreeze proteins on ice nucleation. Proc Natl Acad Sci USA 2016, 113, 14739-14744. doi:10.1073/pnas.1614379114 Meister, K.; DeVries, A.; Bakker, Huib, J.B.; Drori, R. Antifreeze Glycoproteins Bind Irreversibly to Ice. J Am Chem Soc 2018, 140, 9365-9368. doi:10.1021/jacs.8b04966 Omedi, J.O.; Huang, W.; Zhang, B.; Li, Z.; Zheng, J. Advances in present‐day frozen dough technology and its improver and novel biotech ingredients development trends—A review. Cereal Chemistry 2019, 96, 34-56. doi.org/10.1002/cche.10122 Wilkins, L.E.; Hasan, M.; Fayter, A.E.R.; Biggs, C.; Walker, M.; Gibson, M.I. Site-specific conjugation of antifreeze proteins onto polymer-stabilized nanoparticles. Polym Chem 2019, 10, 2986–2990. doi: 10.1039/c8py01719k Voets, I.K. From ice-binding proteins to bio-inspired antifreeze materials. Soft Matter, 2017, 13, 4808-4823, doi: 10.1039/c6sm02867e

3)     In the abstract or introduction part, the author should state clearly the aim of the article. For example, if the review is to summarize the application of IBPs in different industries, the introduction part should be focused on this topic, rather than describing the cold adaptation of organisms in details which almost takes two paragraphs.

-> Thank you for this suggestion; we added a paragraph about the aim and special focus on the industrial applications to the Introduction part:

„The discovery of ice binding proteins (IBPs) synthesized by cold–adapted microorganisms as a strategy of overcoming low temperatures resulted in an increasing interest in their specificity. Their properties such as lowering the freezing point of water or protection from recrystallization during storage opened up a prospect of IBPs becoming useful tools in commercial applications. Special attention is paid to antifreeze proteins (AFPs), which were found to be beneficial biomolecules for the food industry [8], agriculture [9], cryosurgery and cryopreservation of cells [10], tissues [11] and organs [12].

This review describes the biodiversity of IBPs in terms of their structures, roles, sources of origin and mechanisms of action. The main focus of the article is concentrated on the already existing and potential industrial applications of this fascinating group of biomolecules.”

4)      Actually there are quite a few very nice reviews about IBPs in recent years, accounting for their structure-function relationships, action mechanisms, or application in cryopreservation. The present M.S. should address its specific focus which is different from the other reviews, and guide to other reviews if necessary.

(1)     Many relevant papers that shape the understanding of IBPs action mechanisms are missed in the present M.S., indicating a lack of comprehensive research of the author to this area. The following papers are some works that should not be omitted, e.g., Proceedings of the National Academy of Sciences 2016, 113 (51), 14739-14744; J Am Chem Soc 2018, 140 (30), 9365-9368; Proceedings of the National Academy of Sciences of America 2016, 113 (14), 3740-3745;Proceedings of the National Academy of Sciences of the United States of America 2016, 113 (14), 3714-3716.

-> Thank you for this comment, we reviewed the recent articles [89, 90, 99, 100] and extended the Paragraph 4.1. about the mechanisms of action of AFPs. We have also added the citations to review papers from the last 4 years that focus on the mechanisms, structural diversity, methods of activities’ determination and applications of IBPs, e.g.:

Bredow, M.; Walker, V.K. Ice–Binding Proteins in Plants. Front Plant Sci. 2017, 8, 2153. doi: 10.3389/fpls.2017.02153 Omedi, J.O.; Huang, W.; Zhang, B.; Li, Z.; Zheng, J. Advances in present‐day frozen dough technology and its improver and novel biotech ingredients development trends—A review. Cereal Chemistry 2019, 96, 34-56. doi:10.1002/cche.10122 Voets, I. K. From ice-binding proteins to bio-inspired antifreeze materials Soft Matter, 2017, 13, 4808-4823, doi: 10.1039/c6sm02867e

The cover letter is also attached in the online submission system.

Reviewer 3 Report

This review is very useful and provides an excellent summary of the potential of biological cryoprotectants. Bringing together the underlying principles of this area, as far as they are known, and coupling these with the potential applications of ice active proteins is interesting and very helpful.   One area that should be discussed is the development of molecules that are chemically tractable and are based on what has been learnt from studying biological molecules. For example, the section describing the treatment of aluminium with AFP could well lead to developing compounds that mimic the AFP activity and that are more suited to use in commercial settings. Mike Gibson's work on PVA derivatives might be helpful here.   I found the manuscript generally well-written and the structure was excellent but there are some areas where the sentence construction is a bit ambiguous and would benefit from clarification.   L 1 …: I'm not sure there is evidence for life below the crust of the earth so this paragraph could be amended a little to include references to life within clouds, in the outer crust where water is found as well as the areas that are mentioned   L 53: Consider 'The formation of ice crystals can damage …'   L 57–63: It would be helpful here to distinguish between behavioural avoidance where freezing temperatures are evaded, and freezing avoidance itself where organisms are supercooled without ice formation. Examples of the latter include many fish and arthropods such as springtails. At present, the manuscript doesn't seem to distinguish them apart.   L 71–75: Some discussion of supercooling would help clarify this section. Organisms that live close to their freezing point may benefit from supercooling rather than managing ice formation and even in organisms that do freeze, most maintain the cell interior in liquid form suggesting that freezing tolerance and avoidance can coexist (and may have different mechanisms) within the same system.   Figure 1: What are the bases for the relationships shown here (TH>IRI etc.)? A table showing the data here would be helpful. The data in Table 1 have some relationship but they don't really distinguish between IRI and TH in a way that helps understand what is proposed in Figure 1. Quantifying IRI is difficult and so this information might be hard to collate, but it would be useful to know more.   A case can be made for discarding the category of AFP and considering THP and IRIP as separate categories. Most TH proteins seem to have IRI activity, but many plant compounds show IRI but not TH suggesting the underlying mechanism may be different. This might simply be a discussion of terms but the distinction between the two might also be significant in assessing their roles in different aspects of cryoprotection.   L 97: 'The first discovery of AFPs was by Devries et al over 50 years …'   L 98: 'millimolar'   L 110–111: This sentence needs rewriting   L 116–117: It is worth noting the diverse origins of the genes for AFGP in Antarctic fish and northern cod, particularly the convergence of function from quite different genetic sources.   Figure 2: The structure of Marinomons IBP would be useful if added here even though it is theoretical as the way water is organised is very relevant to discussions of IBP mechanisms (Vance, T. D. R., Olijve, L. L. C., Campbell, R. L., Voets, I. K., Davies, P. L., & Guo, S. (2014). Ca2+-stabilized adhesin helps an Antarctic bacterium reach out and bind ice. Bioscience reports, 34(4), 357-368.)   E: the structure shown here is probably a crystallographic dimer and this representation suggests it is functional as a dimer and also implies the residues thought to interact with ice instead are involved with dimerisation. Including just one molecule from the asymmetric unit would be clearer.   L 141–147: see comments at L116–117 above   L 165: this is a very old reference (31) and could be supplemented with something more recent.   L 225–227: I found this a bit confusing   L 243: 'Garnham'   Table 1: Rhodotorula: 'King George'   Tenebrio: I understood mealworms (Tenebrio molitor) to be entirely associated with human activities and I don't think the reference here (64) provides evidence to the contrary. Furthermore, Tenebrio larvae don't seem to survive freezing and it is not clear that their THP really functions in freezing and may play a role instead as a stabilising agent in salt solutions in the malpighian tubules (Ramsay, J. A. (1964). The rectal complex of the mealworm Tenebrio molitor, L. (Coleoptera, Tenebrionidae). Philosophical Transactions of the Royal Society B: Biological Sciences, 248(748), 279-314.). L 264: A reference to this would be useful here (reference 18 might be appropriate, but also Celik, Y., Graham, L. A., Mok, Y. F., Bar, M., Davies, P. L., & Braslavsky, I. (2010). Superheating of ice crystals in antifreeze protein solutions. Proc. Natl. Acad. Sci. U.S.A., 107(12), 5423-5428 and Knight, C. A., & DeVries, A. L. (1989). Melting inhibition and superheating of ice by an antifreeze glycopeptide. Science, 245(4917), 505-507.)   L 287–289: This section needs more explanation to tie the techniques used to what was being investigated. As it stands, I found this hard to follow.   L 312–314: I could not find the evidence for biologically significant dimers in Rhagium from the reference cited and thought that the MALS data included there supported a monomer in solution.   L 335–365: The INP mentioned here are broadly related and may reflect horizontal gene transfer among microorganisms. However, other INP activity is known and recent work on ice algae suggest other structures may play a similar role in other lineages (Bar-Dolev, M., Braslavsky, I., & Davies, P. L. (2016). Ice-Binding Proteins and Their Function. Annual Review of Biochemistry, 85(1), 515-542.).   L 339–340: It might be worth distinguishing between microorganisms that live at low temperatures from those that survive. These are fundamentally different problems and the mechanisms for managing each state may be quite different. This broadly maps to the psychrophile-psychrotolerant distinction but is worth specifically discussing   L 448: What qualities are lost?   L 535: Where did the concentration of chlorophyll increase? Inside the diatoms?   L 544–547: This section is hard to follow and could be rewritten.   L 591–592: I found this sentence confusing.   L 604: Elaborate on what Snowmax and what it does.   L 609: A reference here would be useful.

Author Response

Dear Reviewer,

We would like to gladly thank you for a very good overall score and your beneficial advices. We agree with the suggestions and believe that the changes introduced after your review will undoubtedly increase the quality of the manuscript. Please find our detailed responses to the particular comments below.

Comments and Suggestions for Authors

This review is very useful and provides an excellent summary of the potential of biological cryoprotectants. Bringing together the underlying principles of this area, as far as they are known, and coupling these with the potential applications of ice active proteins is interesting and very helpful.   One area that should be discussed is the development of molecules that are chemically tractable and are based on what has been learnt from studying biological molecules. For example, the section describing the treatment of aluminium with AFP could well lead to developing compounds that mimic the AFP activity and that are more suited to use in commercial settings. Mike Gibson's work on PVA derivatives might be helpful here.

-> Thank you for this suggestion. Indeed this is an important point to raise it in this discussion. We have added a paragraph:

“Finally, Wilkins et al., 2019, have recently demonstrated that AFPs can be conjugated onto polymer-stabilized gold nanoparticles. This may lead to the application of AFPs organized into complex assemblies in colloid science or in basic sciences utilizing the nanoparticle cores for tracking and bioimaging [134]. Moreover, researchers seek for a development of compounds that mimic AFP activity, for example small molecules, polymeric analogues (e.g. polyvinyl alcohol derivatives) or inexpensive metallic salts such as zirconium (IV) acetate [135]. Such materials could increase the commercial availability of antifreeze products, as AFPs are prone to denaturation and expensive to produce.”

and cited two new references:

Wilkins, L.E.; Hasan, M.; Fayter, A.E.R.; Biggs, C.; Walker, M.; Gibson, M.I. Site-specific conjugation of antifreeze proteins onto polymer-stabilized nanoparticles. Polym Chem 2019, 10, 2986–2990. doi: 10.1039/c8py01719k Voets, I.K. From ice-binding proteins to bio-inspired antifreeze materials. Soft Matter, 2017, 13, 4808-4823, doi: 10.1039/c6sm02867e

I found the manuscript generally well-written and the structure was excellent but there are some areas where the sentence construction is a bit ambiguous and would benefit from clarification.  

L 1 …: I'm not sure there is evidence for life below the crust of the earth so this paragraph could be amended a little to include references to life within clouds, in the outer crust where water is found as well as the areas that are mentioned  

-> The paragraph was amended according to the suggestions and 3 relevant references were cited:

Drake, H.; Ivarsson, M.; Bengtson, S.; Heim, C.; Siljeström, S.; Whitehouse, M.J.; Broman, C.; Belivanova, V.; Åström, M.E. Anaerobic consortia of fungi and sulfate reducing bacteria in deep granite fractures. Nat Commun 2017, 8, 55. doi:10.1038/s41467-017-00094-6 Mason, O.U.; Nakagawa, T.; Rosner, M.; Van Nostrand, J.D.; Zhou, J.; Maruyama, A.; Fisk, M.R.; Giovanonni, S.J. First Investigation of the Microbiology of the Deepest Layer of Ocean Crust. PLoS ONE 2010, 5, e15399. doi:10.1371/journal.pone.0015399 Fröhlich-Nowoisky, J.; Kampf, C.J.; Weber, B.; Huffman, A.; Pöhlker, C.; Andreae, M.O.; Lang-Yona, N.; Burrows, S.M.; Gunthe, S.S.; Elbert, W.; Su, H.; Hoor, P.; Thines, E.; Hoffmann, T.; Després, V.R.; Pösch, U. Bioaerosols in the Earth system: Climate, health, and ecosystem interactions. Atmospheric Research 2016, 182, 346-376. doi: 10.1016/j.atmosres.2016.07.018

L 53: Consider 'The formation of ice crystals can damage …'

-> The sentence was rephrased.

L 57–63: It would be helpful here to distinguish between behavioural avoidance where freezing temperatures are evaded, and freezing avoidance itself where organisms are supercooled without ice formation. Examples of the latter include many fish and arthropods such as springtails. At present, the manuscript doesn't seem to distinguish them apart.

-> Thank you for this precious advice, indeed we realised this topic could be changed. We have expanded the Paragraph 2.1. and added the following reference:

[13] Costanzo, J.P.; Lee, R.E. Avoidance and tolerance of freezing in ectothermic vertebrates. J Exp Biol 2013, 216, 1961-1967. doi:10.1242/jeb.070268

L 71–75: Some discussion of supercooling would help clarify this section. Organisms that live close to their freezing point may benefit from supercooling rather than managing ice formation and even in organisms that do freeze, most maintain the cell interior in liquid form suggesting that freezing tolerance and avoidance can coexist (and may have different mechanisms) within the same system.  

-> We have extended the Paragraph 2.1. with the information about supercooling:

“Moreover, supercooling inhibits the formation of ice within the tissue by ice nucleation and allows the cells to maintain water in a liquid state-cell solutions are kept between the equilibrium freezing point and the homogeneous ice nucleation temperature of water (usually between -1 and -41 °C). As a consequence, the water within the cell stays separate from the extracellular ice.  In contrast, tissues that avoid stress survive freezing temperatures by supercooling, a process in which cell solutions are kept between the equilibrium freezing point and the homogeneous ice nucleation temperature of water (usually between -1 and -41°C) [13]. To sum up, the interplay between FA, FT and utilization of supercooling is not straightforward, as FT and FA may coexist within the same system, based on different mechanisms. “

Costanzo, J.P.; Lee, R.E. Avoidance and tolerance of freezing in ectothermic vertebrates. J Exp Biol 2013, 216, 1961-1967. doi:10.1242/jeb.070268

Figure 1: What are the bases for the relationships shown here (TH>IRI etc.)? A table showing the data here would be helpful. The data in Table 1 have some relationship but they don't really distinguish between IRI and TH in a way that helps understand what is proposed in Figure 1. Quantifying IRI is difficult and so this information might be hard to collate, but it would be useful to know more.   A case can be made for discarding the category of AFP and considering THP and IRIP as separate categories. Most TH proteins seem to have IRI activity, but many plant compounds show IRI but not TH suggesting the underlying mechanism may be different. This might simply be a discussion of terms but the distinction between the two might also be significant in assessing their roles in different aspects of cryoprotection.  

-> We changed the composition of the Figure 1 and added the panel 1D. We included the examples of TH values in different organisms producing AFPs to underline the lowering of TH activity. The explanation that some organisms display high TH, but low/no IRI activity (fish, insects), other – moderate TH and IRI activities (microorganisms), and plants exhibit high IRI/low TH activity, was added to description of the Figure 1. IRI values cannot be measured, but we added the sentence about the techniques used to describe it:

„There are techniques used in evaluation of IRI activity, e.g. splat assay, sucrose sandwich splat assay, capillary assay [87].”

Sharma, B.; Deswal, R. Antifreeze Proteins in Plants: An overview with an insight into the detection techniques including nanobiotechnology. Journal of Proteins and Proteomics2014, 5. 89-107.

L 97: 'The first discovery of AFPs was by Devries et al over 50 years …'  

-> The sentence was rephrased.

L 98: 'millimolar'  

-> The word was changed.

L 110–111: This sentence needs rewriting  

-> The sentence was rephrased.

L 116–117: It is worth noting the diverse origins of the genes for AFGP in Antarctic fish and northern cod, particularly the convergence of function from quite different genetic sources.

-> We commented on this situation in L141-147 and expanded the Paragraph 3.1. about AFGPs.

Figure 2: The structure of Marinomons IBP would be useful if added here even though it is theoretical as the way water is organised is very relevant to discussions of IBP mechanisms (Vance, T. D. R., Olijve, L. L. C., Campbell, R. L., Voets, I. K., Davies, P. L., & Guo, S. (2014). Ca2+-stabilized adhesin helps an Antarctic bacterium reach out and bind ice. Bioscience reports, 34(4), 357-368.)   E: the structure shown here is probably a crystallographic dimer and this representation suggests it is functional as a dimer and also implies the residues thought to interact with ice instead are involved with dimerisation. Including just one molecule from the asymmetric unit would be clearer.  

-> Thank you for this suggestion. We added two panels depicting the structure of MpAFP to the Figure 2 (2J-K) and a description of MpAFP’s mechanism of binding to ice (the end of the Paragraph 3):

„In contrast, RIV ice-binding domain of a large microbial adhesin MpAFP does not follow this fold and is stabilized by regularly arranged Ca2+ ions (Figure 2J) [17]. Also RII of MpAFP, which serves a spacer domain projecting RIV away from other cell surface molecules, is stabilized by numerous Ca2+ ions that strengthen and extend the massive array of RII domains to form a rigid rod-like structure (Figure 2K) [18].”

[17] Garnham, C.P.; Campbell, R.L.; Davies, P.L. Anchored clathrate waters bind antifreeze proteins to ice.(2011) Proc Natl Acad Sci USA 2011, 108, 7363-7367. doi: 10.1073/pnas.1100429108

[18] Vance, T. D. R., Olijve, L. L. C., Campbell, R. L., Voets, I. K., Davies, P. L., & Guo, S. Ca2+-stabilized adhesin helps an Antarctic bacterium reach out and bind ice. Bioscience reports 2014, 34, 357-36. doi:10.1042/BSR20140083

L 141–147: see comments at L116–117 above

-> We extended the Paragraph 3.1. about AFGPs:

“The AFGP gene derives from the recruitment and iteration of a small region spanning the boundary between the first intron and second exon of the trypsinogen gene. This new segment was expanded and then iteratively duplicated to produce 41 tandemly repeated segments [38].”

L 165: this is a very old reference (31) and could be supplemented with something more recent.  

-> The more recent reference was added [27].

L 225–227: I found this a bit confusing  

-> The sentences were rephrased.

L 243: 'Garnham'  

-> The surname was changed.

Table 1: Rhodotorula: 'King George'  

-> We corrected the typo.’

Tenebrio: I understood mealworms (Tenebrio molitor) to be entirely associated with human activities and I don't think the reference here (64) provides evidence to the contrary. Furthermore, Tenebrio larvae don't seem to survive freezing and it is not clear that their THP really functions in freezing and may play a role instead as a stabilising agent in salt solutions in the malpighian tubules (Ramsay, J. A. (1964). The rectal complex of the mealworm Tenebrio molitor, L. (Coleoptera, Tenebrionidae). Philosophical Transactions of the Royal Society B: Biological Sciences, 248(748), 279-314.).

-> The origin of the organism was changed in Table 1 to „areas associated with human activites”. Indeed, the reference (64 in the reviewed paper) is not suitable for the origin of these mealworms - it referred to the molecular mass and the activity of mealworm’s AFP.

Indeed, Tenebrio larvae do not seem to survive freezing, but can survive 4 °C during 4 weeks – in these conditions, the larvae were reared in the laboratory to produce high amounts of AFPs (Tomalty. H.E.; Graham, L.A.; Eves. R.; Gruneberg A.K.; Davies, P.L. Laboratory-Scale Isolation of Insect Antifreeze Protein for Cryobiology. Biomolecules 2019, 9, 180; doi:10.3390/biom9050180).

L 264: A reference to this would be useful here (reference 18 might be appropriate, but also Celik, Y., Graham, L. A., Mok, Y. F., Bar, M., Davies, P. L., & Braslavsky, I. (2010). Superheating of ice crystals in antifreeze protein solutions. Proc. Natl. Acad. Sci. U.S.A., 107(12), 5423-5428 and Knight, C. A., & DeVries, A. L. (1989). Melting inhibition and superheating of ice by an antifreeze glycopeptide. Science, 245(4917), 505-507.)  

-> The reference of Celik et al., 2010, was cited as [80].

L 287–289: This section needs more explanation to tie the techniques used to what was being investigated. As it stands, I found this hard to follow.  

-> We expanded the Paragraph 4.1. with the techniques used in the estimation of IRI activity, e.g. splat assay, sucrose sandwich splat assay, capillary assay. We added the citation of the publication in which the authors describe the techniques for measurement of AFP activity and their advantages and disadvantages:

[87] Sharma, B.; Deswal, R. Antifreeze Proteins in Plants: An overview with an insight into the detection techniques including nanobiotechnology. Journal of Proteins and Proteomics 2014, 5. 89-107. DOI

L 312–314: I could not find the evidence for biologically significant dimers in Rhagium from the reference cited and thought that the MALS data included there supported a monomer in solution.  

-> The sentence was rephrased to underline that RiAFP does not form stable dimers in the solution:

„AFP from the longhorn beetle (Rhagium inquisitor, RiAFP) was crystallized as a dimer which, however, was not stable in solution because of higher shape complementarity for RiAFP and ice than for two subunits of RiAFP’s crystallographic dimer [28].”

[28] Hakim, A.; Nguyen, J.B.; Basu, K.; Zhu, D.F.; Thakral, D.; Davies, P.L.; Isaacs, F. J.; Modis, Y.; Meng, W. Crystal Structure of an Insect Antifreeze Protein and its Implications for Ice Binding. J BiolChem2013, 288(17), 12295–12304. doi:10.1074/jbc.M113.450973

L 335–365: The INP mentioned here are broadly related and may reflect horizontal gene transfer among microorganisms. However, other INP activity is known and recent work on ice algae suggest other structures may play a similar role in other lineages (Bar-Dolev, M., Braslavsky, I., & Davies, P. L. (2016). Ice-Binding Proteins and Their Function. Annual Review of Biochemistry, 85(1), 515-542.).  

-> We extended the Paragraph 4.2 with the description of the organisms that also produce INPs and cited the suggested reference [34].

L 339–340: It might be worth distinguishing between microorganisms that live at low temperatures from those that survive. These are fundamentally different problems and the mechanisms for managing each state may be quite different. This broadly maps to the psychrophile-psychrotolerant distinction but is worth specifically discussing.  

-> We reformulated the Paragraph 4.2.:

„Ice nucleation activity is widespread in many Gram negative, pathogenic and epiphytic bacteria. Interestingly, these bacteria can be psychrophiles (living at 0-15 °C) or mesophiles usually living at 30-37 °C but able to survive occasionally at lower temperatures by means of synthesis of e.g. cold-shock proteins (CSP) [76, 101].”

and cited the references:

Kawahara, H. Cryoprotectants and ice–binding proteins, In Psychrophiles: from biodiversity to biotechnology, Margesin R., Schinner F., Marx J. C., Gerday C. (eds), Publisher: Springer–Verlag, Berlin, Germany, 2008, pp. 229–246. Lorv, J.S.H.; Rose, D.R., Glick B.R. Bacterial Ice Crystal Controlling Proteins. Scientifica2014, 976895, 20. doi: 10.1155/2014/976895

and also expanded the Paragraph 2.1. („The strategies can range from behavioral (hibernation under mud, migration to warmer areas, annual lifespans) to physiological including freezing avoiding (FA) by supercooling and freezing tolerance (FT) [13].” and further on).

L 448: What qualities are lost?  

-> We listed the qualities in the text („qualities such as uniform crumb, volume and sensory qualities like odor, taste and crust crispiness [112].”); we also added the reference:

[112] Omedi JO, Huang W, Zhang B, Li Z, Zheng J (2019) Advances in present‐day frozen dough technology and its improver and novel biotech ingredients development trends—A review. Cereal Chemistry 96, 34-56. https://doi.org/10.1002/cche.10122

L 535: Where did the concentration of chlorophyll increase? Inside the diatoms?  

-> The authors of publication [125] mentioned that they measured the concentration by taking one ml of each sample periodically and measured the fluorescence in vivo using a Trilogy laboratory fluorometer. So we believe the concentration of chlorophyll increased inside the diatoms. We added in vivo to the sentence.

L 544–547: This section is hard to follow and could be rewritten.  

-> The paragraph was rewritten. Thank you for helping us to clarify this section.

“Also in 2015, Lee and co–authors published a report where they compared the impact of three different proteins: LeIBP (Glaciozyma sp. AY30), FfIBP (Flavobacterium frigoris) and type III AFP (Zoarces americanus) on cryopreservation of ovaries frozen by vitrification, and their subsequent warming and transplantation into mice. The results showed that supplementing AFPs at a concentration of 20 mg/mL in the vitrification solution had a protective effect on the survival of ovarian tissue during cryopreservation and transplantation, and that the most beneficial activity was observed for the LeIBP [11].”

L 591–592: I found this sentence confusing.  

-> The sentence was rewritten:

„One example where the production of scaffolds and porous materials may be used is the transport of drugs and nutrients through controlled unidirectional freezes with subsequent lyophilization.”

L 604: Elaborate on what Snowmax and what it does.  

-> We added a sentence describing the product („Protein extract from P. syringae works as snow inducer and improves the crystallization proces [137].”) and a reference to the manufacturer’s website: http://www.snomax.com/product/snomax.html [137].

L 609: A reference here would be useful.

-> We added the reference:

[138] Bae, W.; Mulchandani, A.; Chen, W. Cell surface display of synthetic phytochelatins using ice nucleation protein for enhanced heavy metal bioaccumulation. Journal  of  Inorganic  Biochemistry  2002, 88, 223–227. doi: 10.1016/S0162-0134(01)00392-0

The cover letter is also attached to the online submission system.

Round 2

Reviewer 1 Report

The authors have indeed improved the manuscript in many ways, but also that there are still some mistakes in their corrections:

Line 405: crystallizedas => add space
Line 714: add space before 'Protein '
Line 723: "becomes extremely useful tools " => "can make it a very useful tool, as demonstrated " (this is also partly subjective, but the "s" in "tools" is objectively incorrect)

Reviewer 2 Report

Great improvement can be seen in the revised manuscript.

The paper can be published in its present version, and further review is not needed.